# Unhackable Temporal Rewarding for Scalable Video MLLMs

**En Yu**[1,¶]   **Kangheng Lin**[2,¶]   **Liang Zhao**[3,¶]   **Yana Wei**[4]   **Zining Zhu**[5]   **Haoran Wei**[3]
**Jianjian Sun**[3]   **Zheng Ge**[3]   **Xiangyu Zhang**[3]   **Jingyu Wang**[2†]   **Wenbing Tao**[1†]
[1]Huazhong University of Science and Technology
[2]Beijing University of Posts and Telecommunications     [3]StepFun
[4]Johns Hopkins University     [5]University of Chinese Academy of Sciences
{yuen, wenbingtao}@hust.edu.cn

## Abstract

In the pursuit of superior video-processing MLLMs, we have encountered a perplexing paradox: the *"anti-scaling law"*, where more data and larger models lead to worse performance. This study unmasks the culprit: ***"temporal hacking"***, a phenomenon where models shortcut by fixating on select frames, missing the full video narrative. In this work, we systematically establish a comprehensive theory of temporal hacking, defining it from a *reinforcement learning* perspective, introducing the ***Temporal Perplexity (TPL)*** score to assess this misalignment, and proposing the *Unhackable Temporal **R**ewarding (**UTR**)* framework to mitigate the temporal hacking. Both theoretically and empirically, TPL proves to be a reliable indicator of temporal modeling quality, correlating strongly with frame activation patterns. Extensive experiments reveal that UTR not only counters temporal hacking but significantly elevates video comprehension capabilities. This work not only advances video-AI systems but also illuminates the critical importance of aligning proxy rewards with true objectives in MLLM development. Project page: https://Ahnsun.github.io/UTR/.

## 1   Introduction

The pursuit of artificial intelligence that emulates human-like reasoning has increasingly emphasized the role of System 2 cognitive processes—deliberate (DeepSeek-AI et al., 2025; OpenAI, 2024), structured (GPT-4o, 2024), and temporally-aware reasoning (Yu et al., 2024; Fei et al., 2024)—in advancing multimodal large language models (MLLMs). While early MLLMs like GPT-4V (OpenAI, 2023), LLaVAs (Liu et al., 2024b;a) demonstrated remarkable capabilities in static image understanding, their application to video understanding remains constrained by the inherent complexity of spatiotemporal dynamics, long-range context dependencies, and multimodal alignment. This motivates researchers to develop powerful video MLLMs for the open-source community.

The dominant paradigm in video foundation model construction relies on contrastive (Tong et al., 2022; Feichtenhofer et al., 2022; Wang et al., 2024c) or generative learning (Cheng et al., 2024; Zhang et al., 2024c) from extensive video-text pair datasets. However, recent studies have unveiled a counterintuitive *"anti-scaling law"* phenomenon (Xu et al., 2024). Practically, increased data volume (Wang et al., 2024b) or model parameters (Xu et al., 2024) leads to performance degradation. Our analysis in Figure 2(a) also shows that adding more training data decreases temporal modeling performance due to the dilution of high-quality samples. Further investigation reveals models often infer entire captions from a few key frames, typically just the initial (Figure 2(b)) or last one (Figure 1). This suggests that current methodologies inadvertently promote a form of *shortcut learning*. Critically, this issue resists resolution through mere data and parameter scaling. Such approaches may, in fact, exacerbate the problem.

We propose to reframe this issue through the lens of *reinforcement learning* (RL) (Sutton & Barto, 2018). The generative modeling of MLLMs on video-text pairs can be formulated as a sequential decision-making process where the model's policy aims to maximize the expected reward of gen-

---

[†]Corresponding authors, [¶] Core contribution

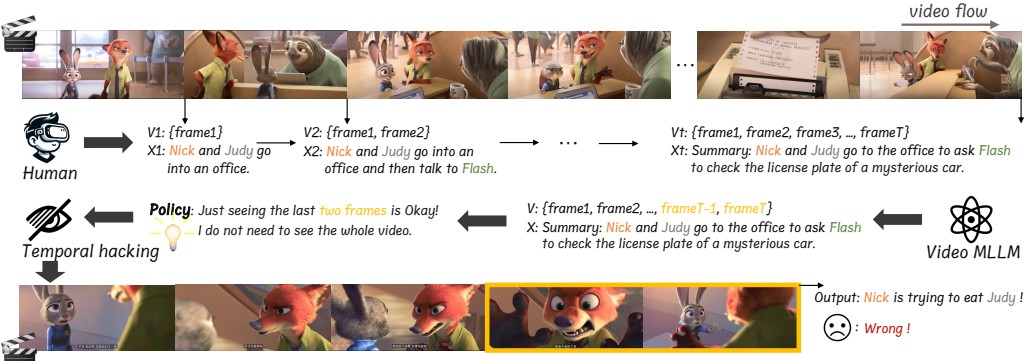

Figure 1: **Illustration of temporal hacking.** We select a scene from the *Zootopia* to vividly illustrate the phenomenon of temporal hacking, where the fox is named *Nick* and the rabbit is named *Judy*. Humans watch videos frame by frame, gradually building an understanding of the content, following a "flow" similar to a Markov process. In contrast, MLLMs process the entire video and its content at once, which can cause them to take shortcuts by focusing only on the most relevant frames.

erating highly relevant text conditioned on video frame context. This formulation necessitates a critical examination: *Does our proxy reward function (video-text or video-caption pair) adequately approximate the true reward (video-language alignment) we aim to optimize?* Empirical evidence suggests a significant misalignment. We observe a manifestation of reward hacking (Skalse et al., 2022) — termed "***temporal hacking***" in the context of video LLMs. This predicament mirrors a boat in a racing game, furiously spinning in circles to collect "power-ups" while never advancing towards the finish line (Jack & Dario, 2016).

Escaping the vortex of temporal reward hacking requires a shift in strategy, not merely increased effort. That is, ***employing a more suitable proxy reward is key*** to overcoming this challenge. To this end, we first investigate the causes of temporal reward hacking and introduce a novel metric, *Temporal Perplexity (TPL)* score, to quantify its severity. Experiments reveal a striking correlation between TP scores and models' temporal modeling capabilities, with higher TPL scores consistently associated with the activation of more video frames. Our analysis further leads to the proposal of two key principles for designing an effective proxy reward function for video MLLMs: *high frame information density* and *high inter-frame information dynamics*. Guided by these two principles, we further propose an *Unhackable Temporal Reward (UTR)*. UTR leverages *spatiotemporal attributes* and *bidirectional queries* to model video-language alignment. Comprehensive experiments validate that UTR, as an automated and scalable method, effectively achieves unhackable temporal modeling by guiding the model's observational tendencies across all frames.

Our contributions are threefold:

- We provide a novel RL perspective on the video MLLM unscaling phenomenon, systematically establishing *"temporal hacking"* theory as its first comprehensive explanation.

- We design the *Temporal Perplexity (TPL)* score, and through extensive experiments, TPL has demonstrated a high correlation with the true performance of the model, providing a reliable reference metric for mitigating temporal hacking.

- Through a series of theoretical and experimental analyses, we propose *two principles* to guide the design of proxy rewards for video-language modeling and further propose *Unhackable Temporal Rewarding (UTR)*. Extensive experiments and analyses substantiate the effectiveness of UTR, offering crucial insights into video MLLM temporal modeling.

## 2 BACKGROUND & EXAMPLE ANALYSIS

### 2.1 WHAT IS TEMPORAL HACKING?

**Reward hacking** (Skalse et al., 2022; Yuan et al., 2019), also known as reward exploitation or reward gaming, refers to a phenomenon in reinforcement learning (RL) where an agent discovers a

way to maximize its reward signal without actually achieving the intended goal of the task designer. Specifically, we first define a sequential decision problem $M = (S, A, P, R, \gamma)$, typically formalized as a Markov decision process (MDP), where $S$ is the state space, $A$ is the action space, $P : S \times A \times S \to [0, 1]$ is the transition probability function, $R : S \times A \times S \to \mathbb{R}$ is the reward function, and $\gamma \in [0, 1]$ is the discount factor. The goal of RL is to find a policy $\pi : S \to A$ that maximizes the expected cumulative discounted reward:

$$J(\pi) = \mathbb{E}_{\tau \sim \pi} \left[ \sum\nolimits_{t=0}^{\infty} \gamma^t R(s_t, a_t) \right],$$
$$\pi^* = \arg\max_{\pi} \mathbb{E}_{\tau \sim \pi} \left[ \sum\nolimits_{t=0}^{\infty} \gamma^t R(s_t, a_t) \right], \tag{1}$$

where $\tau = (s_0, a_0, s_1, a_1, ...)$ is a trajectory generated by following policy $\pi$. $\pi^*$ is the optimal policy obtained under the current reward function. Reward hacking occurs when there exists a policy $\pi_h$ (generally $\pi_h = \pi^*$) such that:

$$J(\pi_h) > J(\hat{\pi}), \quad however, \quad K(\pi_h) \ll K(\hat{\pi}), \tag{2}$$

where $\hat{\pi}$ is the optimal policy for achieving the intended task, and $K$ denotes the true performance of the policy model in the intended task. In essence, reward hacking indicates an optimization misalignment, leading to policies that achieve high proxy rewards ($J(\pi_h)$) but fail to accomplish the true reward objectives ($K(\pi_h)$).

**From reward hacking to temporal hacking.** Autoregressive video-language modeling (Li et al., 2023a; Zhang et al., 2023; 2024c), aims to replicate human video comprehension. As illustrated in Figure 1, humans sequentially access each video frame, incrementally building an understanding by integrating all prior information (Coltheart, 1980). Similarly, the model progressively generates tokens for each frame with the preceding video context conditioned. It is natural to represent this task as a sequential Markov decision process from an RL perspective.

Particularly, given a video frame sequence $V = \{v_t\}_{t=1}^T$ (where $T$ is the total number of frames) and a specific time step $t$, the sequence of preceding frames $V_{1:t}$ constitutes the state space, and the corresponding text token $x_t$ forms the action space. During training, the policy $\pi$ sequentially generates tokens $x_t$ conditioned on state $V_{1:t}$. The generated tokens' quality and relevance to $V_{1:t}$ are evaluated by a reward function $R$, typically measured through the next token's cross-entropy (Radford et al., 2018; 2019). The objective can be formalized as:

$$J(\pi) = \mathbb{E}_{\tau \sim \pi} \left[ \sum\nolimits_{t=1}^{T} \gamma^t R(V_{1:t}, x_t) \right]. \tag{3}$$

By optimizing the policy model based on this objective function $J$, we obtain an optimal policy model $\pi^*$ under the current reward function. However, as shown in Figure 1 and previous works (Xu et al., 2024; Wang et al., 2024b), $\pi^*$ often fails to generate text that accurately aligns with video content and user instructions. Instead, the model may optimize the objective by *accessing only a limited number of frames*, leading to shortcut learning. This issue, termed *temporal hacking* in this paper, reflects the discrepancy between proxy and true objectives as described by Eq. 2.

We provide an illustrative example in Figure 1, where it can be observed that the model, through temporal hacking, has identified a "simpler" version of the true reward by focusing only on the last two frames of the video. This learned proxy reward can be highly dangerous in certain situations, leading to completely erroneous video understanding.

## 2.2 WHAT CAUSES TEMPORAL HACKING?

In this section, we will analyze the causes of temporal hacking phenomenon in video-language modeling from both theoretical and experimental perspectives.

**Theoretical perspectives.** In reward hacking theory (Skalse et al., 2022; Yuan et al., 2019), misalignment between proxy and true objectives ($J(\pi) \neq K(\pi)$) leads to shortcut learning. For video-language modeling, the true objective is to generate spatially and temporally comprehensive descriptions that align with human understanding of the video. However, in practice, the surrogate objective rewards consistency between model predictions and human-annotated captions (Wang et al., 2024b) or curated internet content (Bain et al., 2021; Wang et al., 2023). This discrepancy can result in suboptimal model behavior.

Ideally, as illustrated in Eq. 3, trajectories $\tau = (V_{1:1}, x_1, ..., V_{1:t}, x_t, ...)$ propagates along every frame in the temporal sequence, implicitly necessitating a textual descriptions comprehensively de-

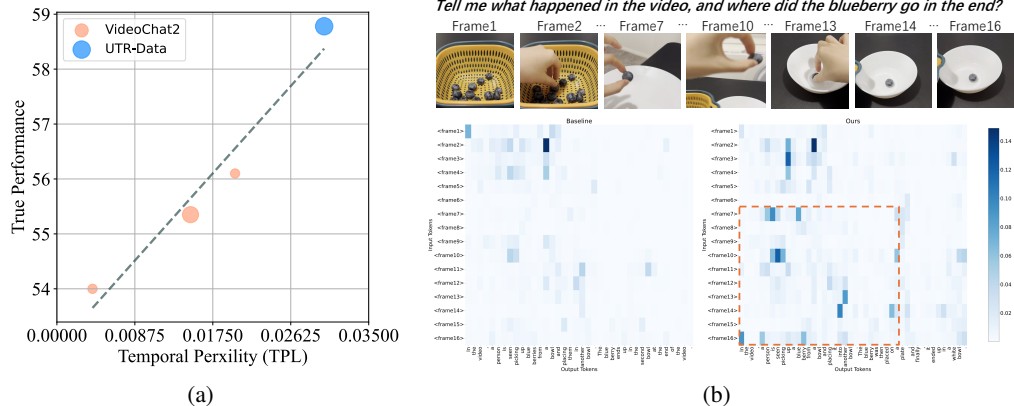

(a)  (b)

Figure 2: **Analysis of the temporal hacking.** **(a)** shows the relationship between temporal perplexity and true performance. The size of the radius of the circle represents the amount of data. **(b)** visualizes the attention map illustrating which specific frames the model's output focuses on.

scribe each frame. However, due to frame redundancy and annotation costs, the text is often conditioned only on a subset of frames or aggregated information from multiple frames, especially in some *static* or *low-motion* scenarios. It is particularly challenging to provide a distinct description for each frame. Consequently, the policy's trajectory becomes $\tau = (V_{1:1}, x_1, ..., V_{k:t}, x_t, ...)$ where $V_{k:t}$ represents any frame set satisfying description $x_t$, and is a subset of $V_{1:t}$. The resultant surrogate objective can be expressed as:

$$J(\pi) = \mathbb{E}_{\tau \sim \pi} \left[ \sum_{t=1,1 \leq k \leq t}^{T} \gamma^t R(V_{k:t}, x_t) \right]. \tag{4}$$

As illustrated in Figure 1, optimizing such a proxy is insufficient and prone to deviate from the true objective of comprehensive video understanding. This temporal hacking can be quantified by subtracting Eq. 3 from Eq. 4, yielding $\Delta \mathcal{R}$:

$$\Delta \mathcal{R} = \sum_{t=1,1 \leq k \leq t}^{T} \gamma^t \left( R(V_{1:t}, x_t) - R(V_{k:t}, x_t) \right). \tag{5}$$

From Eq. 5, it is evident that as $t$ increases, or as the average subset size $k$ increases (indicating that video descriptions can be condensed to fewer frames), the reward gap widens. This elucidates the observed "anti-scaling law" phenomenon in existing video-language models, where performance degrades as video length increases.

**Experimental perspectives.** To shed light on temporal hacking, we propose an extreme perspective to probe $\Delta \mathcal{R}$. We leverage perplexity (Li et al., 2024d) $\mathcal{R}_{ppl}$ to model the cumulative reward between video context and its textual description. Higher similarity correlates with greater cumulative reward and lower model perplexity. We simulate the true cumulative reward using a fully sampled video sequence as video context. To model an extreme case of proxy cumulative reward, we use a single, randomly sampled keyframe to represent the entire video context (*i.e.*, $k = t$). This simulates a scenario where the model attempts to describe the whole video based on minimal information. The difference between these two rewards is defined in this paper as *temporal perplexity (TPL, defined as $\mathcal{T}_{tpl}$)* or *temporal hackability*. Formally,

$$\mathcal{T}_{tpl} = - \left( \mathcal{R}_{ppl}(V_{1:T}, x_T) - \mathcal{R}_{ppl}(V_{T:T}, x_T) \right). \tag{6}$$

In practice, to avoid distributional shift, we utilize our own MLLM model, trained on the full set of video data, to calculate perplexity. We record the mean negative log-likelihood (NLL) loss across all text tokens for each sample (*i.e.*, the logarithm of perplexity) to represent $R_{ppl}$.

By combining Eq. 5 and Eq. 6, we can intuitively infer that, under the same training setup, a lower TPL score indicates a larger $\Delta R$, which in turn leads to a more severe occurrence of temporal hacking. To prove this, we conduct two experiments as shown in Figure 2 for in-depth analysis of the relation between TPL score and temporal hacking.

Specifically, we first fine-tuned models using subsets from VideoChat2 (Li et al., 2024c) data with varying $\mathcal{T}_{tpl}$ ranges and then mixed the data with different TPL. Intuitively, *higher average TPL scores indicate a reduced likelihood of reward hacking, thereby leading to superior video comprehension performance.* Figure 2(a) corroborates this, showing a significant correlation between video

performance and TPL scores across multiple benchmarks, indicating that temporal perplexity effectively measures $\Delta\mathcal{R}$ and even reward hacking. Furthermore, we can also observe that when the TPL score is low, increasing the amount of data does not lead to performance gains, indicating the occurrence of the anti-scaling law phenomenon.

Then we delved deeper by analyzing attention maps of models on identical video-text pairs. Figure 2(b) illustrates that models trained on data with higher average-$\mathcal{T}_{tpl}$ activate more frames during inference on these well-described data. Conversely, models with lower-$\mathcal{T}_{tpl}$, due to severe reward hacking and inferior video modeling, activate fewer frames. These experiments demonstrate that our TPL score can effectively reflect the extent of temporal hacking, providing a reliable metric for exploring strategies to address this issue.

## 3    UNHACKABLE TEMPORAL REWARDING

### 3.1    HOW TO MITIGATE TEMPORAL HACKING?

Section 2 introduces, defines, and analyzes the concept of *temporal hacking*. A novel metric, *temporal perplexity (TPL score)*, is proposed to assess whether the issue of temporal hacking arises in video-language modeling. At this point, the next important question arises: *How can temporal hacking be mitigated or prevented?* Building upon the aforementioned analysis, we first propose two principles to guide the design of an **unhackable** reward in video-language temporal modeling:

   ***Principle I: High frame information density.*** *The content of the video text should uniquely correspond to as many frames as possible.*

   ***Principle II: High inter-frame information dynamics.*** *Descriptions for different frames should be coherent and reflect temporal variations and event progression.*

The Principle I, as delineated by Eq. 5, aims to mitigate the $\Delta\mathcal{R}$ by reducing $k$ as discussed in Section 2.2. This can be accomplished by ensuring each frame of the video is uniquely described. The Principle II emphasizes continuous dynamics, not only to further reduce $k$ and $\Delta\mathcal{R}$, but also to ensure the continuity of policy state transitions in Eq. 3, thereby enhancing the model's understanding of real-world physical laws.

Current temporal modeling approaches predominantly focus on maximizing the relevance and consistency of video information (Principle II). However, addressing Principle I remains challenging due to high frame rates and inter-frame redundancy, complicating textual descriptions of individual frames. Advanced techniques such as InternVID (Wang et al., 2023) and COSMO (Wang et al., 2024a) ameliorate information density to some extent through video interleave formats, yet they still struggle with the high information density of frames and fail to effectively model spatiotemporal dynamics, thus not fully addressing Principle II. Additionally, methods like COSA (Chen et al., 2023b), which concatenate image-text pairs to create video data, fail to establish spatiotemporal relationships between frames, entirely violating Principle II.

To simultaneously satisfy the two proposed principles, we further propose the *Unhackable Temporal Rewarding (UTR)* to boostrap the video-language modeling.

### 3.2    UNHACKABLE TEMPORAL REWARDING

As validated in Section 2, suboptimal proxy rewards easily lead to temporal hacking in models. To address this, we propose a novel temporal rewarding method adhering to the aforementioned principles. Our approach, illustrated in Figure 3, extracts spatiotemporal attributes from video frames (row 1) and uniformly queries them (row 2) to model video-language alignment. This automated and scalable method achieves unhackable temporal modeling by guiding the model's observational tendencies across all frames.

**Spatiotemporal attributes are key to representing unique video frame content.** Mitigating temporal hacking is challenging due to high frame rates and information redundancy in videos as mentioned before. We propose extracting *spatiotemporal attributes* (e.g., trajectory, identity, action) to capture relatively independent information from each frame. This approach offers two advantages:

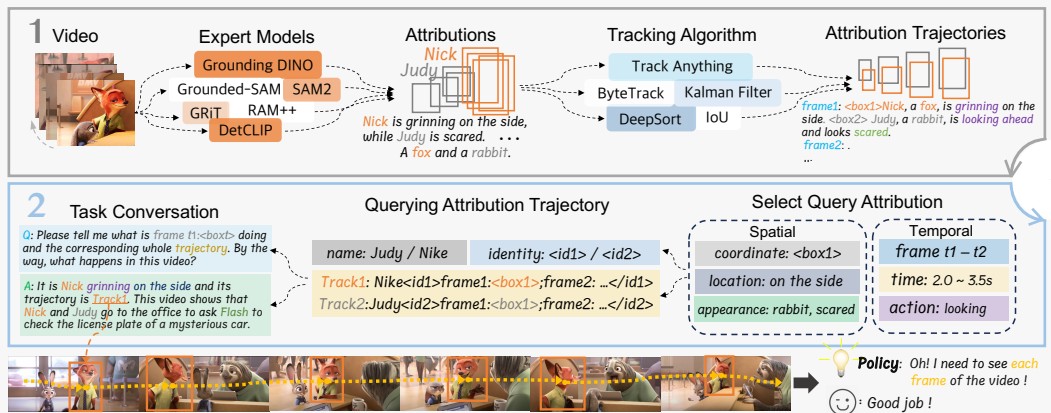

Figure 3: **Overall pipeline of Unhackable Temporal Rewarding (UTR)**. UTR begins by using a mixture of expert models to extract unique spatiotemporal attributes and employs a tracking algorithm to construct multiple subject trajectories based on confidence levels (data modeling, top). It then performs bidirectional querying of temporal and spatial attributes to generate dialogue data (task modeling, bottom), thereby learning spatiotemporal dynamics.

- *Frame-to-frame variations in attributes, especially positional coordinates, enable modeling of frame-specific information, increasing information density (aligning with **Principle I**).*
- *These attributes function as queries to link information across the video, facilitating learning of spatiotemporal dynamics (aligning with **Principle II**).*

Specifically, given a video frame sequence $V = \{v_t\}_{t=1}^{T}$ with the same meaning in Eq. 3, we extract the attribute information of subjects from each frame as follows:

$$X_t = \{x_t^{loc}, x_t^{app}, x_t^{act}\} = F(v_t), \tag{7}$$

where $x_t^{loc}, x_t^{app}, x_t^{act}$ indicate the location, appearance, and action information of subjects in frame $v_t$, respectively. Function $F$ extracts this information, using labeled data or specialized models such as GRiT (Wu et al., 2022) and Grounding DINO (Liu et al., 2023a). We then organize this subject information into trajectories corresponding to each subject:

$$\{Y_i\}_{i=1}^{N} = \left\{\{y_{i,t}^{tr}, y_{i,t}^{id}, y_{i,t}^{act}\}_{t=1}^{T}\right\}_{i=1}^{N} = A(\{X_t\}_{t=1}^{T}), \tag{8}$$

where $Y_i$ is the trajectory of subject $i$ and $N$ is the number of subjects in the video. To be specific, $y_{i,t}^{tr}, y_{i,t}^{id}, y_{i,t}^{act}$ indicate the trajectory, identity, and action information of subject $i$ in frame $v_t$, respectively. Function $A$ associates subjects across frames to form trajectories and identities, typically using tracking algorithms like ByteTrack (Zhang et al., 2022).

**Bidirectional querying explicitly models spatiotemporal dynamics.** Previous methods (Wang et al., 2023; 2024a) modeled relatively dense information by interleaving text with selected frames, yet they neglected the critical spatiotemporal dynamics. Inspired by Merlin (Yu et al., 2024), we propose a bidirectional querying mechanism that uses any temporal or spatial attribute to query global spatiotemporal attributes. This approach offers two benefits:

- *Explicit modeling of spatiotemporal attributes forces the model to read each frame, aligning with **Principle I**.*
- *The arbitrariness of querying across time and space enhances the model's understanding of spatiotemporal dynamics, and the stronger this arbitrariness, the deeper the understanding, aligning with **Principle II**.*

Particularly, we randomly sample the information of one or more subjects as query attributes and select several frames as query frames. The model must predict the complete subject information based on the provided query data. Formally,

$$P(Y|V, Y_q) \sim P\left(\{y_{s_i}\}_{i=1}^{N} | \{v_t\}_{t=1}^{T}, \{y_{s_i, t_j}\}_{i,j}\right), \tag{9}$$

where $\{s_i\}_{i=1}^{N} \subseteq \{1, 2, \ldots, N\}$ represents sampled subject identities, $\{t_j\}_{j=1}^{M} \subseteq \{1, 2, \ldots, T\}$ indicates selected query frames, and $y_{s_i, t_j}$ denotes attribute information of selected subjects sampled from $Y$, which can be location, appearance, and action description.

Notably, the random selection of query frames $\{t_j\}_{j=1}^M$ ensures the model utilizes query information from any part of the video—beginning, middle, or end—as cues to trace the entire trajectory. This approach not only compels the model to fully observe and comprehend the entire video, avoiding shortcuts like relying solely on initial or final frames, but also enhances its understanding of time-dependent physical laws. By necessitating the model to infer states across various temporal intervals, it implicitly learns concepts such as momentum, velocity, and acceleration, thereby strengthening its grasp of fundamental spatiotemporal dynamics.

## 4 EMPIRICAL RESULT DETAILS

### 4.1 EXPERIMENT SETTINGS

**Datasets.** We primarily construct UTR-Data using several existing open-source video datasets, namely HowTo100M (Miech et al., 2019), MeViS (Ding et al., 2023), and LaMOT (Li et al., 2024e). To extract subject attributions from each video frame, we use the region-to-text detector GRiT (Wu et al., 2022). Subsequently, we apply the ByteTrack (Zhang et al., 2022) tracking algorithm to construct attribution trajectories. Further details can be found in the Appendix B.

**Implementation Details.** To apply our UTR modeling strategy within the current video MLLM, we have developed a novel video MLLM, *i.e.*, ***Video-UTR***. For the specific Video-UTR pipeline, we follow the general architecture in LLaVA-NeXT-Video (Zhang et al., 2024c), which consists of a vision encoder, SigLIP-L (Zhai et al., 2023), a large language model, QWen-2 (Yang et al., 2024), and a modality alignment projector, 2-layer GeLU-MLP. The training process consists of two stages. (1) *Stage I*: Modality alignment, where only the projector is trained using the $558K$ LLaVA (Liu et al., 2024b) dataset. (2) *Stage II*: Multi-task joint training, where the LLM is trained with various task datasets including video instruction-following data. Here, we mainly apply our ***UTR*** in the *Stage II*, which combines the constructed task data based on UTR with LLaVA-NeXT SFT data. Further details about the training settings can be found in the Appendix B.

### 4.2 GENERAL COMPREHENSION EVALUATION

To showcase the generality and effectiveness of the proposed paradigm, we evaluated Video-UTR across various understanding benchmarks. Using the standard MLLM evaluation framework and the LLMs-Eval tool (Zhang et al., 2024a), we assessed major image and video understanding tasks. Results are shown in Tables 1 and 2. For video understanding, we focused on three general benchmarks: MVBench (Li et al., 2024c), TempCompass (Liu et al., 2024c), and VideoMME (Fu et al., 2024), as well as four video QA benchmarks: MVSD-QA (Xu et al., 2017), MSRVTT-QA (Xu et al., 2016), TGIF-QA (Jang et al., 2017), and ActivityNet-QA (Caba Heilbron et al., 2015). For image understanding, we reported scores from popular benchmarks like MM-Vet (Yu et al., 2023), MMBench (Liu et al., 2023b), MMMU (Yue et al., 2024), MME (Fu et al., 2023), LLaVA-wild (Liu et al., 2024b), SEED (Ge et al., 2023b), AI2D (Kembhavi et al., 2016), and RealWorldQA (xAI, 2024). For fairness, we used results from original papers.

**Video Understanding.** Table 1 shows that Video-UTR outperforms other video MLLMs on most benchmarks, ranking first in 4 out of 7 tasks, highlighting its strong video understanding capabilities. Its high scores on MVBench (58.78%), TempCompass (59.67%), and VideoMME (52.63%) demonstrate its ability to handle complex tasks like temporal reasoning, identifying differences, locating objects, tracking motion, and interpreting dynamic scenes. Additionally, its performance on four video QA benchmarks reflects exceptional understanding, particularly in managing temporally sensitive information. Remarkably, Video-UTR achieves these results using only about $1.1M$ video samples, a much smaller dataset compared to other models of similar performance, showcasing the efficiency and effectiveness of our UTR approach.

**Image Understanding.** Table 2 shows that Video-UTR, despite being a video MLLM, delivers highly competitive performance compared to image-level MLLMs. For instance, on key benchmarks, Video-UTR matches or outperforms top image MLLMs like LLaVA-1.5 (Liu et al., 2024b) (39.6% vs. 35.4% on MM-Vet) and the stronger LLaVA-NeXT-Img (Zhang et al., 2024c) (76.6% vs. 72.1% on MMBench). It also performs well on hallucination benchmarks, achieving 88.9% on POPE, and excels in image QA, with 63.7% on RealWorldQA, showing its ability to avoid misidentification and misalignment with irrelevant image details. These results demonstrate that UTR not

Table 1: **General Video Understanding Performance Comparsion** on 7 benchmarks, Video-UTR outperforms competitors in 4 out of 7 benchmarks and ranks second on the others, despite these competitors using larger training datasets or more parameters. Several benchmark names are abbreviated due to space limits. TempC: Tempcompass, ANet-QA: ActivityNet-QA. And Acc indicates Accuracy. The best results are **bold** and the second-best results are underlined. ∗ indicates metrics reproduced by ourselves for evaluation cause original paper does not report.

| Methods | LLM | Data Scale | MVBench | TempC | VideoMME | MSVD-QA | | MSRVVT-QA | | TGIF-QA | | ANet-QA | |
|---|---|---|---|---|---|---|---|---|---|---|---|---|---|
| | | | | | | Acc | Score | Acc | Score | Acc | Score | Acc | Score |
| VideoChat (2023a) | Vicuna-7B | 765K | 35.5 | – | – | 56.3 | 2.8 | 45.0 | 2.5 | 34.4 | 2.3 | – | 2.2 |
| VideoChat2 (2024c) | Vicuna-7B | 1.9M | 51.1 | 38.5 | – | 70.0 | 3.9 | 54.1 | 3.3 | – | – | 49.1 | 3.3 |
| Video-ChatGPT (2023) | Vicuna-7B | 765K | 32.7 | 31.8 | – | 51.6 | 2.5 | 29.6 | 1.8 | – | – | 12.4 | 1.1 |
| Video-LLaVA (2023) | Vicuna-7B | 765K | 34.1 | 34.8 | 39.9 | 64.9 | 3.3 | 49.3 | 2.8 | 51.4 | 3.0 | 35.2 | 2.7 |
| VideoLLaMA2 (2024) | LLaMA2-7B | 13.4M | 54.6 | – | 46.6 | 70.9 | 3.8 | – | – | – | – | 50.2 | 3.3 |
| PLLaVA (2024) | LLaMA2-7B | 1M | 46.6 | – | – | 76.6 | 4.1 | 62.0 | 3.5 | 77.5 | 4.1 | 56.3 | 3.5 |
| LLaVA-NeXT-Video (2024c) | Qwen2-7B | 860K | 54.6 | – | 33.7 | 67.8 | 3.5 | – | – | – | – | 53.5 | 3.2 |
| LLaVA-OneVision(2024a) | Qwen2-7B | 1.6M | 56.7 | 59.0* | 58.2 | 65.3* | 3.8* | 43.3* | 3.0* | 52.8* | 3.4* | 56.6* | 3.3* |
| Video-UTR (**Ours**) | Qwen2-7B | 1.1M | **58.8** | **59.7** | 52.6 | 73.5 | 4.1 | 58.3 | 3.6 | 56.4 | 3.6 | 55.0 | 3.2 |

Table 2: **General Image Understanding Performance Comparision** on 9 benchmarks, Video-UTR achieves performance comparable to, or even surpassing, that of pure image-level MLLMs. LLaVA[W]: LLaVA in the wild. The best results are **bold** and the second-best results are underlined.

| Methods | LLM | MM-Vet | MMBench | MMMU | MME | LLaVA[W] | POPE | SEED | AI2D | RealWorldQA |
|---|---|---|---|---|---|---|---|---|---|---|
| *Image-level MLLM* | | | | | | | | | | |
| InstructBLIP (2024) | Vicuna-7B | 33.1 | 36.0 | 30.6 | 1137.1 | 59.8 | 86.1 | 53.4 | 40.6 | 36.9 |
| Qwen-VL-Chat (2023b) | Qwen-7B | **47.3** | 60.6 | 37.0 | 1467.8 | 67.7 | 74.9 | 58.2 | 63.0 | 49.3 |
| LLaVA-v1.5-7B (2024a) | Vicuna-7B | 30.5 | 64.3 | 35.7 | 1510.7 | 61.8 | 86.1 | 58.6 | 55.5 | 54.8 |
| LLaVA-v1.5-13B | Vicuna-13B | 35.4 | 67.7 | 37.0 | 1531.3 | 66.1 | 88.4 | 61.6 | 61.1 | 55.3 |
| ShareGPT4V (2023a) | Vicuna-7B | 37.6 | 68.8 | 37.2 | 1567.4 | 72.6 | 86.6 | 69.7 | 58.0 | 54.9 |
| LLaVA-NeXT-Img (2024c) | LLaMA3-8B | 44.4 | 72.1 | 41.7 | 1551.5 | 63.1 | 87.1 | – | 71.6 | 60.0 |
| *Video-level MLLM* | | | | | | | | | | |
| LLaMA-VID (2023b) | Vicuna-7B | – | 66.6 | – | 1521.4 | – | 86.0 | 59.9 | – | – |
| Video-LLaVA (2023) | Vicuna-7B | 32.0 | 60.9 | – | – | 73.1 | 84.4 | – | – | – |
| LLaVA-NeXT-Video (2024c) | QWen2-7B | 42.9 | 74.5 | 42.6 | 1580.1 | **75.9** | 88.7 | 74.6 | 71.9 | 60.1 |
| Video-UTR (**Ours**) | Qwen2-7B | 39.6 | 76.6 | 43.4 | 1583.6 | 69.4 | 88.9 | 74.7 | 72.1 | 63.7 |

only helps video MLLMs overcome temporal hacking but also enhances their ability to analyze and understand images effectively.

## 4.3 ABLATION STUDY ABOUT UTR

**Effectiveness of each component of UTR.** In this ablation study, we evaluate the impact of removing the two key components of UTR: data modeling (*UTR-Data*) and task modeling (Bidirectional Querying) from Video-UTR. We focus on three major video and image understanding benchmarks. As shown in Table 3, removing UTR-Data and Bidirectional Querying leads to a significant drop in performance on video understanding tasks, emphasizing their importance in handling complex video reasoning tasks. Notably, removing UTR-Data causes a more consistent and pronounced decline across all benchmarks, including both image and video tasks. This underscores the critical role of *data modeling* in UTR, as it directly aligns with the *two principles* we proposed.

At the same time, to eliminate the potential influence of video data volume, we also add an equivalent amount of VideoChat2 (Li et al., 2024c) data. It can be observed that the additional video data did not result in further gains, which further underscores the importance of data modeling. Video data constructed in an improper manner will inevitably lead to temporal hacking, thus hindering the improvement of true video understanding performance.

**Ablation on the scalability of Video-UTR.** In Section 2.2, we identify that the "anti-scaling law" phenomenon observed in current video MLLMs is due to the issue of temporal hacking. To address this, we propose UTR as a mitigation strategy. In this experiment, we will demonstrate whether video MLLMs, with the integration of UTR, can exhibit scalability. As shown in Table 4 and Table 5, thanks to the incorporation of UTR, Video-UTR demonstrates a certain degree of scalability in

Table 3: **Abalation study of Video-UTR** on both video and image understanding benchmarks.

| Ablation Setting | Data Scale | MVBench | TGIF-QA | ANet-QA | MMVet | MMBench | POPE |
|---|---|---|---|---|---|---|---|
| Video-UTR | 1.1M | **58.78** | **56.44** | **55.00** | 39.59 | **76.63** | 88.86 |
| - Task Modeling | 1.0M | 58.45 | 56.11 | 54.21 | 37.33 | 76.37 | **89.29** |
| - Data Modeling | 780K | 54.63 | 54.74 | 54.15 | **42.20** | 75.77 | 89.13 |
| + More VideoChat2 | 1.1M | 57.65 | 53.39 | 53.65 | 36.56 | 75.95 | 88.76 |

Table 4: **Scalability of video data size.**

| UTR-Data Size | MVBench | TempCompass | VideoMME |
|---|---|---|---|
| 0K | 54.63 | 58.88 | **53.37** |
| 180K | 58.45 | 58.47 | 52.30 |
| 325K | **58.78** | **59.67** | 52.63 |

Table 5: **Scalability of frame length.**

| Frame Length | MVBench | TempCompass | VideoMME |
|---|---|---|---|
| 8 | 50.93 | **56.28** | 52.56 |
| 24 | 50.08 | 56.14 | **52.81** |
| 32 | **51.40** | 56.11 | 52.07 |

the size of video data. Specifically, the larger the volume of video data, the better the model's performance. Additionally, we observe that under the condition of unchanged video content, an increased number of video frames does not negatively impact the model's performance. This scalability is advantageous for further exploring the better performance of Video-UTR in the future.

## 4.4 SPATIAL-TEMPORAL UNDERSTANDING OF VIDEO-UTR

Spatial and temporal comprehension are equally important for multimodal video understanding. Here, we evaluate Video-UTR's performance in these two areas using the latest benchmark, MM-ID, in a zero-shot setting. MM-ID tests a model's ability to recognize identities across four increasingly complex levels, focusing on matching and locating objects with different identities across frames. As shown in Table 6, Video-UTR achieved highly competitive scores on both the matching and location sub-metrics without any MM-ID training data. Moreover, it outperformed methods using significantly larger datasets, further demonstrating the strength of the UTR approach. By leveraging spatiotemporal attribute modeling, UTR effectively enables the model to learn both spatial and temporal aspects.

Table 6: **Zero-shot spatial-temporal understanding performance on MM-ID** (Ji et al., 2024).

| Methods | Matching ↑ | Location ↑ | Q&A ↑ | Caption ↑ |
|---|---|---|---|---|
| *Open-source Models* | | | | |
| MMICL (2023a) | – | – | 3.53 | 3.18 |
| SEED (2023a) | – | – | 3.19 | 3.58 |
| QwenVL-Chat (2023b) | – | 0.504 | 3.63 | 2.65 |
| InternLM-XComposer2 (2024b) | – | 0.106 | 3.44 | 2.93 |
| *Closed-source APIs* | | | | |
| QwenVL-Plus (2023b) | 0.313 | 0.187 | 3.87 | 3.79 |
| QwenVL-Max (2023b) | 0.224 | 0.301 | 4.64 | 4.23 |
| Gemini-pro (2023) | 0.687 | 0.081 | 4.97 | 4.04 |
| GPT-4V (2023) | 0.627 | 0.244 | 4.77 | 4.67 |
| Video-UTR (**Ours**) | 0.277 | 0.328 | 4.36 | 3.62 |

Table 7: **Zero-shot long-range video understanding performance on MMBench-Video** (Fang et al., 2024).

| Methods | Overall ↑ | Perception ↑ | Reasoning ↑ |
|---|---|---|---|
| Claude-3.5-Sonnet (2024) | 1.35 | 1.4 | 1.04 |
| VideoChat2-HD (2024c) | 1.23 | 0.44 | 1.23 |
| PLLaVA-34B (2024) | 1.16 | 1 | 1.1 |
| LLaVA-NeXT-Video-34B-HF (2024c) | 1.13 | 0.58 | 1.03 |
| Video-UTR-7B (**Ours**) | 1.35 | 1.38 | 1.24 |

In addition, we rigorously assess the performance of our Video-UTR on the recently introduced and exceptionally demanding long-range video understanding benchmark, MVBench-Video. As illustrated in Table 7, our Video-UTR exhibits remarkable competitiveness on the performance of video perception and reasoning, surpassing a multitude of models with 34B parameters and even larger architectures, despite its compact 7B parameter size.

## 4.5 IN-DEPTH ANALYSIS ABOUT THE TPL SCORE

In Section 2.2, we design a novel metric, temporal perplexity (TPL score), to measure the alignment degree between the proxy reward and the true reward. In this part, we aim to elucidate the correlation between TP scores and true rewards more intuitively from the perspective of video data quality. Specifically, we randomly select 100 video-text pairs from WebVid (Bain et al., 2021) and calculate their temporal perplexity based on the definition in Eq. 6. Then we pick two representative examples to illustrate the relationship between temporal perplexity (TPL) and the quality of video-text pairs.

As shown in Figure 4, it can be observed that *higher TPL score indicates a higher information density in the video or a more detailed description*. In this scenario, the model struggles to describe the entire video using just a single frame. Conversely, if the model's performance based on a single frame is nearly as good as when using all frames, it either suggests that the video's dynamics are

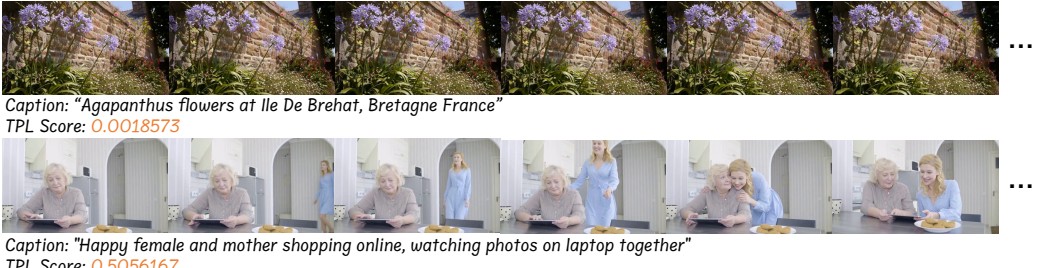

Caption: *"Agapanthus flowers at Ile De Brehat, Bretagne France"*
TPL Score: *0.0018573*

Caption: *"Happy female and mother shopping online, watching photos on laptop together"*
TPL Score: *0.5056167*

Figure 4: **Quantitative comparison** of the video-text pair with different temporal perplexity

negligible, making it almost like an image, or the textual description is so sparse that additional video information does not significantly improve modeling. The result aligns with our discussion in Section 2.2 and the analysis in Figure 2. This case study demonstrates that the TPL score can serve as a useful metric for filtering high-quality video-text pair data. Please refer to Appendix C for more in-depth investigation. Furthermore, for a more comprehensive and in-depth analysis and discussion of the UTR methodology, please refer to Appendix E.

## 5 RELATED WORK

**Multimodal video foundation models.** Recently, vision-language models (Liu et al., 2024b; Zhu et al., 2023; Zhao et al., 2023b; Wei et al., 2024b;a) have demonstrated versatile visual understanding through visual instruction tuning (Liu et al., 2024b; Zhu et al., 2023). However, real-world video comprehension in multimodal models is still in its early stages. Due to the absence of powerful video encoders in the community, existing mainstream video MLLMs (Zhang et al., 2024c; 2023; Cheng et al., 2024) still rely on established images encoder, *i.e.*, CLIP (Radford et al., 2021), to extract visual information frame by frame. Subsequently, they integrate temporal modeling techniques, *e.g.*, Q-former (Zhang et al., 2023), 3D Conv (Cheng et al., 2024), and Pooling (Zhang et al., 2024c; Xu et al., 2024), to compress the visual tokens before feeding them with language tokens into LLMs.

In addition to advancing the design of powerful temporal modules, recent works have increasingly acknowledged the pivotal role of *video-language modeling* in video comprehension. Some works try to design filtering mechanisms (Wang et al., 2024b) to obtain high-quality video data with fine-grained description, while others aim to construct appropriate data structures (Wang et al., 2023; 2024a; Chen et al., 2023b) and task formats (Yu et al., 2024) to enhance modeling performance. In this work, we systematically present how to design effective video-language modeling from a reinforcement learning perspective and propose guiding principles along with example frameworks.

**Reward hacking theory** was firstly introduced in the field of RL as a special case of Goodhart's Law (Goodhart, 1984; Leike et al., 2018), and later explored in the context of AI alignment (Leike et al., 2017). (DeepMind, 2018) formalizes reward hacking by identifying types of reward mis-specifications that lead to it. Subsequent works (Pan et al., 2022; Laidlaw et al., 2024) try to deal with reward hacking from different aspects. On the other hand, reward hacking is not exclusive to reinforcement learning, it also occurs in the optimization of pretrained visual generation models, where approaches often optimize towards a reward model by directly backpropagating gradients from a differentiable reward model (Li et al., 2024b; Zhang et al., 2024b). In this work, we transfer the concept of reward hacking to video-language modeling and establish a novel *temporal hacking* theory to explain the shortcut learning in the existing video MLLM.

## 6 CONCLUSION

In this work, we propose the theory of ***temporal hacking*** from a reinforcement learning perspective to explain shortcut learning in video MLLMs. We introduce a novel metric, ***Temporal Perplexity (TPL)***, to quantify the severity of temporal hacking. Through extensive experiments, we use the TPL score to analyze the causes and features of temporal hacking, leading to the development of two guiding *principles* for video-language modeling. We further propose *Unhackable Video-Language Modeling (UTR)* and build a powerful video MLLM, *i.e.*, ***Video-UTR***. We hope this work offers a new perspective and insights to help the community build more robust video-AI systems.

## ACKNOWLEDGEMENTS

This work was supported by the National Natural Science Foundation of China under Grant 62176096. This work was also supported by the National Natural Science Foundation of China under Grant 62071067.

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

## A    APPENDIX

In this appendix, we provide additional details about *temporal hacking* and our ***Unhackable Temporal Rewarding (UTR)***, which were omitted due to the 10-page limit of the main paper. Specifically, Section B elaborates on the dataset and training settings of Video-UTR. Section C presents additional experiments to analyze UTR's characteristics. Section D offers more qualitative examples to demonstrate the capabilities of Video-UTR, and Section E provides further discussion of existing approaches.

## B    ADDITIONAL DETAILS ABOUT EXPERIMENTAL SETTING

**Additional information of the datasets.** In Section 3.2 of the manuscript, we introduced how we established the unhackable temporal rewarding (UTR) including *data modeling (UTR-Data) and task modeling (Bidirectional Querying)*. Now, in this section, we go into greater detail about how we collected and built the UTR-Data and how we constructed task conversation. To start, we provide an overview of our collected data in Table 8, and then dive into the step-by-step process of how it was constructed.

Table 8: **Training Data Statistics.** We first build our UTR-Data mainly based on sampled HowTo100M, MeViS, and LaMOT. Then we mix UTR-Data with several existing video conversation data, *i.e.*, LLaVA-NeXT-SFT and VideoChat2.

| Modality | Dataset | Original | Used | Ratio% | Training Stage |
|---|---|---|---|---|---|
| Video-Text | HowTo100M (Miech et al., 2019) | 100M | 50K | 0.05% | Stage II |
| | MeViS (Ding et al., 2023) | 443K | 90K | 20.3% | Stage II |
| | LaMOT (Li et al., 2024e) | 2.44M | 225K | 10.5% | Stage II |
| | VideoChat2 (Li et al., 2024c) | 2M | 100K | 5% | Stage II |
| Image-Text | BLIP-558K (Liu et al., 2024b) | 558K | 558K | 100% | Stage I |
| | LLaVA-NeXT-SFT (Zhang et al., 2024c) | 790K | 790K | 100% | Stage II |
| Vision-Language | Total | 106.231M | 1.813M | 1.71% | Stage I & II |

Specifically, we follow the steps below to pre-process the raw video data to construct UTR-Data

(1) Randomly sample the fixed number (16, 24 or 32) frames at a certain frame (gap = 3,4 or 5) or random interval to form a video clip each time.

(2) Extract all spatiotemporal attribution trajectories containing their category, identity, action and bounding boxes in each video clip. This can be accomplished through expert models, *e.g.*, GRiT (Wu et al., 2022), Grounding DINO (Liu et al., 2023a), and ByteTrack (Zhang et al., 2022) or directly obtained from the annotations provided by datasets.

(3) Remove the trajectory containing too small objects (smaller than $1/32$ of the image size).

(4) Random select observation (spatial or temporal attributions in the randomly selected frame) as the query to conduct bidirectional querying task modeling.

(5) Compose the task format as the following:

***Question:*** *System prompt + query question.*

***Answer:*** *query answer, cat1<idi>Frame1:<box>;Frame2:<box>;...</idi>,*

where `<query question, query answer>` is the question-answer pair that is designed based on the selected querying attributes.

**Additional Training Setting Details.** As stated in the manuscript, Video-UTR follows a two-stage training procedure. In this part, we will provide a detailed overview of our training settings, including the hardware used for training, the duration, and the training hyperparameters. All information are recoderd in Table 9.

Table 9: **Training hyperparameters of Video-UTR**. The hyperparameter placed in the middle indicates that this hyperparameter is used in both stages.

| Configuration | Stage I | Stage II |
|---|---|---|
| Machine | NVIDIA Tesla A800 80GB GPU x 64 | |
| Training hours | 1 hour | 20 hours |
| ViT init. | SigLIP-so400m-patch14-384 | Video-UTR Stage I |
| LLM init. | Qwen2-7B-Instruct | Video-UTR Stage I |
| Projection init. | random | Video-UTR Stage I |
| Image resolution | $384^2$ | $384^2$ |
| ViT sequence length | 2048 | 2048 |
| LLM sequence length | 32K | 32K |
| Video Frame length | 1 | 32 |
| Optimizer | AdamW | |
| Optimizer hyperparameter | $\beta_2 = 0.95, eps = 1e^{-8}$ | |
| Peak learning rate | Vision Tower: $2e^{-6}$; LLM: $1e^{-5}$ | |
| Minimum learning rate | 0 | |
| ViT learning rate decay | 0.9 | 0 |
| ViT Drop path rate | 0 | |
| Learning rate schedule | cosine decay | |
| Weight decay | 0.05 | |
| Gradient clip | 1.0 | |
| Training steps | 1k | 5k |
| Warm-up ratio | 0.003 | 0.003 |
| Global batch size | 512 | 256 |
| Gradient Acc. | 1 | 4 |
| Numerical precision | bfloat16 | |
| Optimizer sharding | ✓ | |
| Activation checkpointing | ✗ | |
| Model parallelism | ✗ | |
| Pipeline parallelism | ✗ | |

Table 10: **Video benchmark evaluation setting.** We report some detailed setting during evaluation. MCQ: Multi-choice question. QA: Question-answer.

| Benchmark | Evaluation type | Prompts | Input frames | Answer selection |
|---|---|---|---|---|
| Temoral Compass | MCQ | `Question` + "Please directly give the best option:" | 32 | GPT score |
| | Yes or No | `Question` + "Please answer yes or no:" | 32 | |
| | Caption Matching | `Question` + "Please directly give the best option:" | 32 | |
| | Captioning | `Question` | 32 | |
| MVBench | Video - MCQ | "Question" + `Question` +"Option:" + `Options` + "Only give the best option." | 32 | Option matching |
| VideoMME | Video-MCQ | These are the frames of a video. Select the best answer to the following multiple-choice question based on the video. Respond with only the letter (A, B, C, or D) of the correct option. | 32 | Option matching |
| MSVD | Video QA | `Question` | 32 | GPT Score |
| ActivityNetQA | Video QA | `Question`+ "Answer the question using a single word or phrase." | 32 | GPT Score |
| TGIFQA | Video QA | `Question` | 32 | GPT Score |
| VideoChatGPT | Video QA | `Question` | 32 | GPT Score |

**Additional Testing Setting Details.** In the inference and evaluation phase, we essentially follow the settings of PLLaVA (Xu et al., 2024) and LLaVA-NeXT-Video (Zhang et al., 2024c), including the system prompt for inference, the number of frames extracted, and so on, while conducting evaluations on the existing video benchmark. Specifically, as illustrated in Table 10, we mainly use the uniform frame sampling for frame selection. For answer selection based on GPT score, we mainly use the gpt-3.5-turbo-0613 version to evaluate the responses of our model.

## C  ADDITIONAL EXPERIMENTAL ANALYSIS

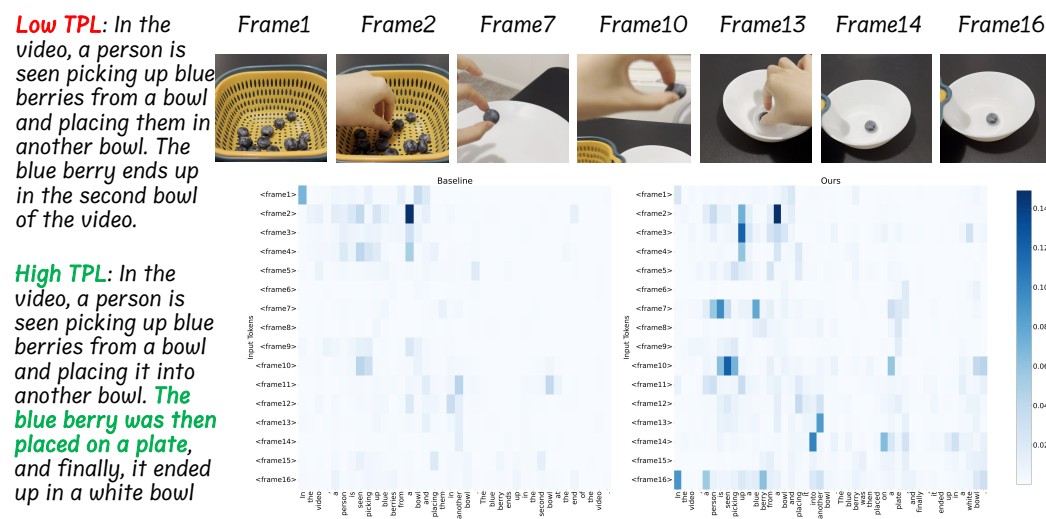

Figure 6: **Output attention visualization.** We compute the average output layer attention of the tokens generated by the model for each frame in the QA task and visualized the results.

**More analysis about temporal perplexity (TPL).** In the Section 4.5, we present a case study to illustrate the relationship between the proposed TPL score and data quality, where a higher TPL score indicates better data quality. In this part, we further present the relationship between TPL and data quality from a quantitative statistical perspective. Specifically, we calculate the TPL score for different data subsets in VideoChat2 (Li et al., 2024c) and computed their average values. The results are shown in Figure 5. We can observe that the TPL distribution for the YouCook2 (Zhou et al., 2018) and TextVR (Wu et al., 2025) subset is relatively high. This suggests that these two data subsets are of relatively high quality. As we know, these datasets, such as YouCook2, contain a large amount of first-person perspective and high-motion video data. These videos are rich in high information density and dynamic content, which is beneficial for the model's temporal modeling. The results further prove that TPL provides a reference for selecting high-quality data from VideoChat2. Based on the TPL distribution, sampling more reasoning data is likely to be more beneficial for achieving better video-language modeling.

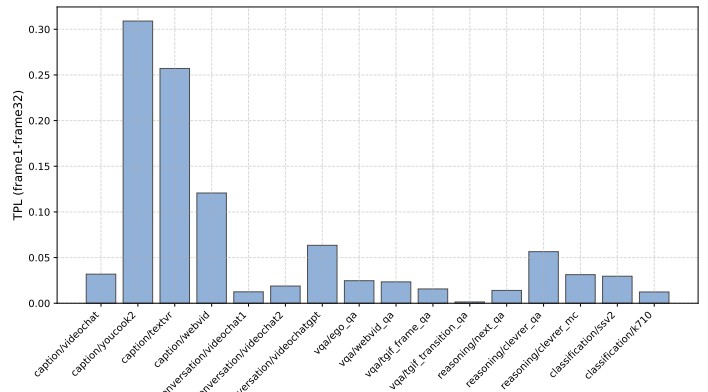

Figure 5: **Quantitative TPL statistic** of VideoChat2.

**More attention visualization analysis.** In Figure 2(b), we present the attention map visualizations of frame tokens under model responses at different TPL levels. In this part, we further provide more detailed attention analysis. As shown in Figure 6 & 7, we conduct two forms of attention visualization. The first involves video QA, visualizing the attention values between the answer content and the tokens of each video frame. The second form calculates the self-attention when inputting the video-text pair into the model simultaneously. From both visualization results, we can observe that our Video-UTR, while achieving a higher TPL score, clearly attends to more frames, thereby avoiding the loss of crucial details in the video and making the answers more accurate and detailed.

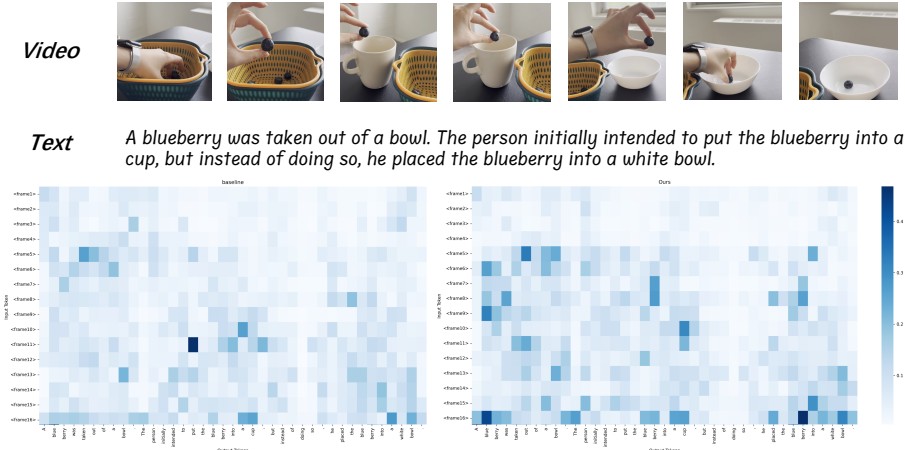

Figure 7: **Video-text input attention visualization.** The left is the attention map of the model with low TPL while the right is the attention map with high TPL score.

## D  ADDITIONAL QUALITATIVE ANALYSIS

In this section, to more intuitively demonstrate the unhackable capability of Video-UTR, we present several subjective video Q&A cases, as shown in Figure 8. Compared to our baseline, LLaVA-Next-Video, our Video-UTR demonstrates a more accurate video understanding capability, specifically by better comprehending user queries, focusing on more video details, and providing more precise and less hallucinated responses. These results further validate the effectiveness of our proposed UTR modeling.

## E  MORE DISSCUSIONS

### E.1  DEPENDENCY ON EXPERT MODELS FOR UTR

As introduced in Section 3.2 of the main text, UTR leverages existing expert models to extract spatiotemporal attribute cues, which serve as the foundation for data modeling and task modeling. Therefore, the capability of the selected expert models and the quality of the extracted attributes are critical variables that significantly influence the effectiveness of UTR modeling. In this part, we will delve into the significance of selecting expert models, the selection criteria, the details of attribute extraction, and the validation of attribute quality.

**The importance and rationale behind selecting expert models for attribute extraction.** The use of expert models to support MLLM training has become a widely adopted strategy in the current development stage. Notable implementations include models such as PaLI-X (Chen et al., 2023c), Qwen-VL (Bai et al., 2023a), InternVL (Chen et al., 2024), and LLaMA3.2-Vision (Meta, 2024), which integrate domain-specific expert models spanning areas like detection, grounding, and OCR to scale up training data annotation. The effectiveness of this approach has been well-validated through extensive empirical studies. Fundamentally, these pipelines operate as a distillation process, transferring knowledge from expert models to MLLMs to enhance specific capabilities, such as fine-grained perception. In line with this paradigm, our proposed UTR framework employs expert models to extract spatiotemporal attributes from video data, thereby strengthening the spatiotemporal perception abilities of video MLLMs. This improvement is substantiated by the empirical results presented in Table 4.4 of our manuscript.

**Extraction and filtering of high-quality attributes.** To select specific expert models, we conducted a systematic evaluation based on existing benchmarks, *e.g.*, COCO (Lin et al., 2014), Lvis (Gupta et al., 2019), VG (Krishna et al., 2017), *etc.*, of the performance of various options, such as GRiT (Wu et al., 2022) and GroundingDINO (Liu et al., 2023a), to identify the most suitable candidates. For the proposed spatiotemporal attributes—including bounding boxes, captions, identities, and actions, as illustrated in Figure 3 of our manuscript—we implemented a multi-stage selection and filtering process. **First**, we filtered the attributes based on the *confidence scores* provided by

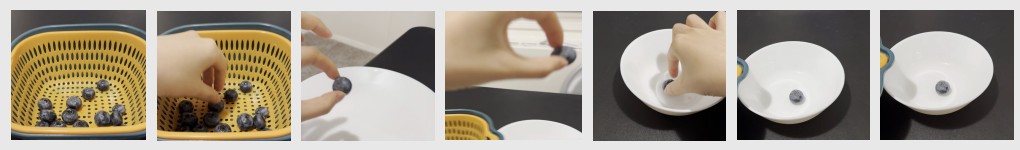

**Human:** Tell me what happened in the video, and where did the blueberry go in the end?

**LLaVA-NeXT-Video:** In the video, a person is seen picking up blue berries from a bowl and placing them in another bowl. The blue berry ends up in the second bowl of the video.

**Ours:** In the video, a person is seen picking up blue berries from a bowl and placing it into another bowl. The blue berry was then placed on a plate, and finally, it ended up in a white bowl

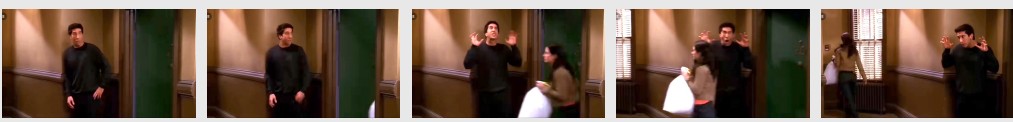

**Human:** Please describe the video in detail.

**LLaVA-NeXT-Video:** The man and woman are standing in a hallway, and the man is holding a white bag. They seem to be having a conversation, but it's unclear what they're saying. The woman is wearing a blue shirt, and the man is wearing a black shirt. There is a green door behind them, and a window on the wall. The man appears to be looking at something off-camera, while the woman looks directly at the camera.

**Ours:** In the video, a man is seen in a hallway and entering a room. He then proceeds to walk towards the woman who is getting out of the room. The man appears to be in a playful mood as he starts making funny faces at the woman, seemingly trying to entertain her. Overall, the video depicts a light-hearted moment between two individuals who seem to be having a good time together.

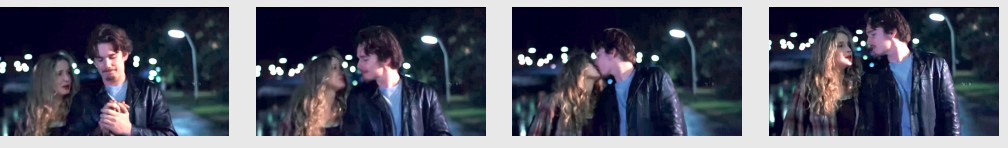

**Human:** Tell me what is the man doing with the woman? Please describe in detail.

**LLaVA-NeXT-Video:** The man and woman are standing in the dark, and the man is holding her hand. They seem to be enjoying each other's company as they stand close together. The video shows them from different angles, but it is clear that they are sharing a romantic moment.

**Ours:** In the video, a man and woman are seen walking down a street at night. The man is wearing a black jacket, and they both appear to be in their late teens or early twenties. They continue walking for a while, with the man occasionally glancing over at her. Eventually, they stop and stand facing each other, and the man leans in to kiss her. The woman reciprocates the kiss, and they stand there for a moment before pulling away from each other. Overall, the video depicts a romantic moment between two young people who seem to have a strong connection.

Figure 8: **Qualitative examples visualization** of Video-UTR. Please note that we only display the most important frames from the full video (32 frames) to conserve space.

the expert models. **Next**, we applied a multi-object tracking algorithm, *i.e.*, ByteTrack (Zhang et al., 2022), to analyze contextual correlations within the video content. This analysis included examining factors such as the Intersection over Union (IoU) of bounding boxes across frames and trajectory continuity metrics, ensuring that trajectory lengths exceeded predefined thresholds. This comprehensive process ensures the reliability and consistency of the extracted attribute trajectories, thereby enhancing their overall quality and utility.

**Human validation of the extracted attributes.** To further validate the effectiveness of the extracted spatiotemporal attributes from video data, we conducted a human evaluation experiment. Specifically, 100 data samples generated using

Table 11: **Human validation** of extracted attributes.

| Validation | Location | Description | Consistency |
|---|---|---|---|
| Human | 2.98 | 2.23 | 2.57 |

our UTR pipeline were randomly selected for assessment by human evaluators. Human annotators will score these data based on three criteria: the accuracy of the subject bounding box, the correctness of the attribute descriptions, and the consistency of the attribute trajectories, using a scoring range of 1 to 3. The results is shown in Table 11. We can observe that the average quality score of the extracted attributes is quite high, indicating a strong level of reliability. The results of this evaluation highlight the robustness and high quality of both the extracted spatiotemporal attributes and the constructed data, confirming the reliability of our pipeline.

### E.2 CONSISTENCY OF TPL WITH HUMAN JUDGMENT

Table 12: **Consistency between TPL score and human judgment**.

| Validation | High | | Medium | | Low | |
|---|---|---|---|---|---|---|
| | Richness | Relevance | Richness | Relevance | Richness | Relevance |
| TPL level | 3 | 3 | 2 | 2 | 1 | 1 |
| Human | 2.85 | 2.76 | 2.15 | 1.85 | 1.61 | 1.64 |

In Section 4.5, we point out that TPL not only reflects the degree of temporal hacking in the video-language modeling process, but it can also serve as a high-order metric to indicate the quality of video-text pairs. In this part, we plan to further explore this issue by examining the consistency of TPL with human judgment, highlighting the reliability of TPL score as a data filtering metric.

Specifically, we first randomly select 100 video-text pairs from VideoChat2 (Li et al., 2023a) and calculate their temporal perplexity based on the definition in Eq. 6. Next, we sort the data by their TPL values and divide it into three groups: high, medium, and low. We then invite several human annotators to rate these sampled video-text pairs on a scale of 1 to 3. The criteria for scoring includes two aspects, *i.e.*, the richness of the video-text information (considering both information density and dynamics) and the relevance of the video to the text. Based on the annotators' scores, the consistency can be evaluated based on the average human ratings and their alignment with the level categories. Before the human annotators begin the annotation process, we provide a detailed annotation guideline, which explains the scoring criteria and standards comprehensively and includes relevant references. Specifically, the two composite metrics, *i.e.*, richness and relevance, are defined as follows:

- **Richness:**
  - Frame information density, referring to the degree to which each frame corresponds to an independent description.
  - Level of descriptive detail, referring to the richness of details included from the video.
  - The richness of motion information, including the extent of motion in both the subject and the scene.
- **Relevance:**
  - The relevance between the video and the text, specifically, the degree to which the description corresponds to the video content.
  - The relevance of the video context, specifically, the extent to which the descriptions of relevant subjects in the video reflect dynamic changes.

Follow the above standards, annotators conducted the validation of the sampled cases and ultimately summarized their assessments into scores for two composite metrics. The results is shown in Table 12. We can observe that the groupings based on TPL scores and those based on human judgments are generally consistent. This indicates that our proposed TPL score is a reliable metric for filtering high-quality video-text pair data.

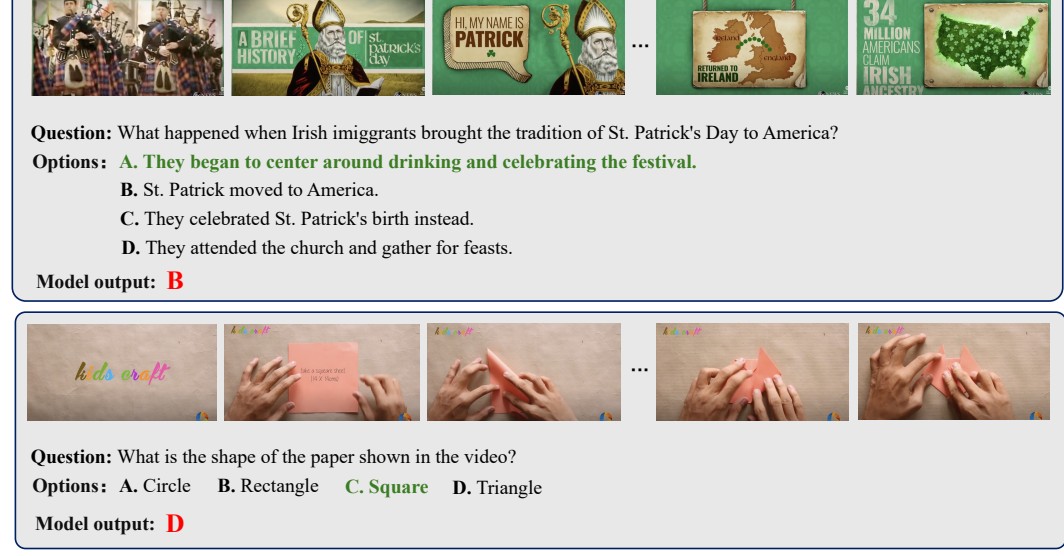

Figure 9: **Failure case visualization** of Video-UTR. We select two representative failure cases from the VideoMME (Fu et al., 2024) benchmark.

### E.3 Failure Case Analysis of UTR

Although our proposed UTR significantly mitigates temporal hacking from both data modeling and task modeling perspectives, it still has limitations in some situation, and we identify several representative examples on the VideoMME (Fu et al., 2024) benchmark. As illustrated in Figure 9, the top case shows that Video-UTR does not perform as well on certain knowledge-oriented Video MCQ tasks. This type of question tests the inherent knowledge base of large language models, so our UTR method does not result in a significant improvement. The bottom case illustrates that in scenarios where the answer can be determined by analyzing a single frame or a few frames, our UTR method does not demonstrate a significant advantage. Placing more emphasis on the overall video content does not provide notable benefits in addressing such questions.

The aforementioned failure cases analysis also highlights the need to design better video understanding benchmarks that can more reasonably and reliably evaluate the ability of video MLLMs to observe and comprehend the overall video content, rather than relying heavily on the inherent capabilities of LLMs.

### E.4 Limitaion and Furture Work.

**Limitation of Unhackable Temporal Hacking.** Although our proposed UTR significantly mitigates temporal hacking from both data modeling and task modeling perspectives, it has a noticeable limitation in terms of its reliance on expert model accuracy. Since UTR modeling is based on extracted subject attributes, the quality of these attributes—such as positional accuracy, precise descriptions of the subject's appearance and actions, and the accuracy of trajectory associations—directly impacts the overall performance of the final model. Therefore, improving the quality of these extracted subject attributes represents a highly valuable direction for future improvement.

**Future work.** On the other hand, seamlessly integrating the constructed attribute trajectories into dialogues poses yet another challenging issue. Exploring whether a single multimodal large language model (Yu et al., 2024; Dong et al., 2024a; Peng et al., 2024) can be utilized to handle the entire data processing and task construction pipeline is a highly promising research direction. In addition, the establishment of a more explicit temporal rewarding mechanism represents a valuable research direction. Leveraging reinforcement learning algorithms, *e.g.*, DPO (Rafailov et al., 2024; Zhu et al., 2025), PPO (Schulman et al., 2017) and GRPO (DeepSeek-AI et al., 2025), for post-training enhancement of the model's video comprehension and reasoning capabilities constitutes a key focus for future research.

