# OpenReview forum: "Unhackable Temporal Reward for Scalable Video MLLMs"
_ICLR.cc/2025/Conference — ICLR 2025 Poster_

### Official Review · Reviewer_ezXT · 2024-11-02

**Soundness:** 3
**Presentation:** 3
**Contribution:** 3
**Rating:** 8
**Confidence:** 3

**Summary:**

The paper introduces Unhackable Temporal Rewarding (UTR), a framework to improve multimodal large language models (MLLMs) by addressing a phenomenon called "temporal hacking". This phenomenon occurs when MLLMs, which process video frames sequentially to understand narratives, exploit shortcuts by focusing on only a few frames (sometimes just the first and the last), thereby missing the entire context. This issue has led to an anti-scaling paradox where increasing model size or data volume degrades performance. The authors propose UTR to counteract this problem by enforcing comprehensive temporal processing, ensuring the model engages with the full video content.

**Strengths:**

The paper is well-written, with the key contributions listed clearly in the introduction. The authors also identified relevant literature in this line of research and highlighted their contributions in the paper. The authors introduce very fundamental principles that should be used to design a video captioning metric, which are high frame information density and high inter-frame information dynamics. They construct a metric by extracting the location, appearance, and action information of subjects in a frame using pre-trained models. They construct datasets based on 3 open-source video datasets, from which they extract subject attributions from each frame in the video.

**Weaknesses:**

1. Since the proposed metric relies on subject attributions extracted from pre-trained models, the errors from those models might propagate into the extracted metadata and subsequent make the metric unreliable. Can the authors comment on whether this is a viable concern?
2. Have the authors measured the reliability of extracted metrics in the presence of multiple subjects in one frame?

**Questions:**

Please see weaknesses above.

---

> ### Author Response · Authors · 2024-11-21
> **Response to ezXT**
>
> We sincerely thank the reviewer for the high evaluation and high rating of our work and the valuable comments and questions provided. We have responded to each of your comments and questions as follows.
>
> - **Q1. Discussion on the quality of the expert model used for attribute extraction.**
>
> 	Thanks for your valuable comments. Yes, the quality of the attributes extracted by the expert model is important to the final performance. We want to elaborate on this from three perspectives.
>
> 	1. **The importance and rationale behind selecting expert models for attribute extraction.** The use of expert models to support MLLM training has become a widely adopted strategy in the current development stage. Notable implementations include models such as PaLI-X [1], Qwen-VL [2], InternVL [3], and LLaMA3.2-Vision [4], which integrate domain-specific expert models spanning areas like detection, grounding, and OCR to scale up training data annotation. The effectiveness of this approach has been well-validated through extensive empirical studies.
>
> 	2. **Extraction and filtering of high-quality attributes.** This paper specifically designs a dedicated data modeling pipeline to extract high-quality spatiotemporal attributes and implements a series of measures to filter out low-quality attributes. Specifically, we first conduct a systematic evaluation based on existing benchmarks, eg., COCO, Lvis, VG, etc., of the performance of various options, such as GRiT and GroundingDINO, to identify the most suitable candidates. For the proposed spatiotemporal attributes—including bounding boxes, captions, identities, and actions, as illustrated in Figure.3 of our manuscript—we implement a multi-stage selection and filtering process. First, we filter the attributes based on the confidence scores predicted by the expert models. Next, we applied a multi-object tracking algorithm, ie., ByteTrack, to analyze contextual correlations within the video content. This analysis included examining factors such as the Intersection over Union (IoU) of bounding boxes across frames and trajectory continuity metrics, ensuring that trajectory lengths exceeded predefined thresholds. This comprehensive process ensures the reliability and consistency of the extracted attribute trajectories, thereby enhancing their overall quality and utility.
>
> 	3. **Human validation of the extracted attributes.** To further confirm that our approach can extract attributes of sufficiently high quality, we conducted an additional human evaluation experiment. Please refer to **Q2** for more details.
>
> 	Please refer to the third common response and Appendix E.1 of the new updated manuscript for more details.
>
> 	[1] Chen et al., PaLi-X: On scaling up a multilingual vision and language model.\
> 	[2] Bai et al., Qwen-vl: A frontier large vision-language model with versatile abilities.\
> 	[3] Chen et al., Internvl: Scaling up vision foundation models and aligning for generic visual-linguistic tasks.\
> 	[4] Meta AI, Llama 3.2: Revolutionizing edge AI and vision with open, customizable models.
>
> - **Q2. The reliability of extracted metrics in the presence of multiple subjects in one frame.**
>
>   We sincerely appreciate your valuable feedback. To better address your concerns, we have incorporated an additional human quality assessment experiment to more accurately evaluate the reliability of our multiple subjects attribution extraction. Specifically, 100 data samples generated using our UTR pipeline are randomly selected for assessment by human evaluators.
>   Human annotators score these data based on three criteria: the accuracy of the subject bounding box, the correctness of the attribute descriptions, and the consistency of the attribute trajectories, using a scoring range of 1 to 3. The results are shown below:
>
>
>   | Validation   | Location | Description | Consistency |
>   |--------------|----------|-------------|-------------|
>   | Human        | 2.98     | 2.23        | 2.57        |
>
>   We can observe that the average quality score of the extracted attributes is quite high, indicating a strong level of reliability. The results of this evaluation highlight the robustness and high quality of both the extracted spatiotemporal attributes and the constructed data, confirming the reliability of our pipeline. Please refer to Appendix E.1 for more details.

---

### Official Review · Reviewer_Edyo · 2024-11-04

**Soundness:** 3
**Presentation:** 3
**Contribution:** 3
**Rating:** 6
**Confidence:** 4

**Summary:**

The paper addresses a paradox observed in training video-processing Multimodal Large Language Models (MLLMs), termed the "anti-scaling law," where increasing data and model size leads to decreased performance. The authors identify "temporal hacking" as the culprit—a phenomenon where models focus on select frames instead of capturing the full video narrative, effectively shortcutting temporal understanding. I believe this behaviour is analogous to posterior collapse in VAEs, where models find ways to ignore part of the input while still optimizing the objective, suggesting a fundamental challenge in enforcing comprehensive information utilization in deep learning models. The authors present a theoretical framework from a reinforcement learning perspective to explain temporal hacking and introduce the Temporal Perplexity (TPL) score as a metric to assess misalignment in temporal modelling. To mitigate temporal hacking, they propose the Unhackable Temporal Rewarding (UTR) framework, which enhances video comprehension by aligning proxy rewards with true objectives through spatiotemporal attribute extraction and bidirectional querying. This solution addresses the root cause: the misalignment between training objectives and desired behaviour in video understanding. Empirical results show that TPL correlates strongly with temporal modelling quality and that UTR significantly improves video MLLMs' performance across multiple benchmarks, while also demonstrating scalability with increased data and frame lengths.

**Strengths:**

- **Originality**: The paper introduces the novel concept of temporal hacking in video MLLMs and frames it within a reinforcement learning context, offering a fresh perspective on the anti-scaling issue.
- **Theoretical foundation**: The reinforcement learning perspective provides clear insights into the problem and guides the development of the solution.
- **Practical metric**: The introduction of the Temporal Perplexity (TPL) score provides a quantifiable measure of temporal misalignment, which correlates with model performance and can guide future research.
- **Proposed solution**: The Unhackable Temporal Rewarding (UTR) framework is a constructive approach to mitigate temporal hacking, showing significant improvements in video comprehension tasks.
- **Comprehensive evaluation**: The paper presents extensive experiments across multiple benchmarks and includes detailed ablation studies, demonstrating the effectiveness of the proposed methods.
- **Practical impact**: The proposed solution is implementable and shows strong empirical results, indicating its potential applicability in real-world video MLLMs.

**Weaknesses:**

- **Clarity and organization**: My main issue is that the paper's presentation suffers from unclear explanations and a lack of cohesion in some sections. Technical terms are sometimes introduced without sufficient context, making it hard to follow, at least to me. I even found Figure 1, which aims to provide a clear illustration of the phenomenon, relatively unclear.
- **Limited discussion of computational overhead**: While the supplementary material provides some training details, the paper lacks a thorough analysis of the computational cost introduced by the UTR framework, particularly regarding the extraction of spatiotemporal attributes using expert models. This makes it difficult to assess the practicality and scalability of the approach compared to existing methods.
- **Dependency on expert models**: The method relies heavily on the accuracy of expert models for attribute extraction, which may limit its applicability if these models are not sufficiently accurate or if they introduce significant computational overhead.
- **Theoretical depth**: The reinforcement learning framework and the concept of temporal hacking could be more rigorously developed. Definitions and theoretical justifications need strengthening.
- **Related work**: There is a lack of thorough comparison with existing methods addressing similar issues, which limits understanding of the paper's novelty and significance in the broader context.
- **Limited exploration of failure cases**: The paper could benefit from more analysis of scenarios where UTR might not perform well, such as with highly redundant or static video content.
- **Reproducibility**: Details on experimental settings, such as hyperparameters, data preprocessing, and model configurations, are insufficient, hindering the ability to replicate results.

**Questions:**

1. Could the authors provide more detailed explanations of the TPL score calculation and its computational overhead? How does it scale with longer video sequences?
2. How does the computational complexity of UTR compare to existing video MLLM approaches? Is the additional overhead from attribute extraction justified by the performance gains?
3. Have the authors explored how the quality and computational cost of expert models affect the final performance? Is there a minimum accuracy threshold needed for these models, and how does their use impact the overall training time?
4. Are there any limitations or potential drawbacks to the UTR approach in scenarios with highly redundant or static video content?
5. Could the authors provide more details about how TPL correlates with human judgment of video understanding quality?
6. Can the framework be applied to other domains involving sequential data, such as speech or time-series analysis?

---

> ### Author Response · Authors · 2024-11-21
> **Response to Reviewer Edyo**
>
> We sincerely appreciate your recognition of our work, as well as your thoughtful and professional review. The suggestions you provided are highly valuable for the further improvement of our work. We have responded to each of your comments and questions as follows.
>
> - **Q1. Issue about clarity and organization of some sections.**
>
> 	Thanks for your suggestions. We have added more detailed captions, descriptions of our methods and more experimental details in the new updated version. Please refer to our new version manuscript.
>
> - **Q2. Discussion about computational overhead.**
>
>   Please refer to the second point of common response. And we can also provide a specific figure about the resources and time required for attribute extraction using expert models.
>
>   | Operation  | Method | Resources      | Time | Data Scale |
>   | ----------- | ----------- | ----------- |----------- | ----------- |
>   | Video frame extraction |  FFmpeg   | 8 * 64 CPU      | 2 days | 7M |
>   | Attribution extraction  | GRiT | 8 * 8 GPU        | 2 days | 7M |
>   | Trajectories construct  | ByteTrack | 8 * 8 GPU        | 3 days | 7M |
>
>
> - **Q3. Discussion about expert model in UTR including dependency, quality and computational cost.**
>
> 	Please refer to the second point of common response, **Q2** and the Appendix of new updated manuscript.
>
> - **Q4. Potential drawbacks in highly redundant or static video content.**
>
> 	We have not observed any limitations in our Video-UTR regarding the understanding of highly redundant or static video content. In fact, as we can see from the results in Table 2 of the manuscript, Video-UTR still demonstrates strong capabilities in image, which can be considered a form of static video to some extent, understanding.
>
> - **Q5. More details about how TPL correlates with human judgment of video understanding quality.**
>
> 	Thank you very much for your valuable and insightful question. To address this issue, we conducted additional human validation experiments, specifically manually ranking video data at different TPL score levels to reflect its consistency with human preferences. Specifically, we first randomly select 100 video-text pairs from VideoChat2 and calculate their temporal perplexity (TPL) based on the definition in Eq.6. Next, we sort the data by their TPL values and divide it into three groups: high, medium, and low. We then invite several human annotators to rate these sampled video-text pairs on a scale of 1 to 3. The criteria for scoring includes two aspects, ie., the richness of the video-text information (considering both information density and dynamics) and the relevance of the video to the text. Based on the annotators’ scores, the consistency can be evaluated based on the average human ratings and their alignment with the level categories.
>
>   | Validation   | High          |               | Medium         |               | Low            |               |
>   |--------------|---------------|---------------|----------------|---------------|----------------|---------------|
>   |              | Richness      | Relevance     | Richness       | Relevance     | Richness       | Relevance     |
>   | TPL level    | 3             | 3             | 2              | 2             | 1              | 1             |
>   | Human        | 2.85          | 2.76          | 2.15           | 1.85          | 1.61           | 1.64          |
>
>
> 	We can observe that the groupings based on TPL scores and those based on human judgments are generally consistent. This indicates that our proposed TPL score is a reliable metric for filtering high-quality video-text pair data. Please refer to Appendix E.2 for more details.
>
> - Q6. **Lack of failure cases, experimental details.**
>
> 	Thanks for your valuable comment. In the newly uploaded version, we dedicate a separate section to discuss failure cases (**Appendix E.3**), specifically analyzing knowledge-based questions that can be answered by observing only a single frame or a few frames. Additionally, we included more experimental details, covering the settings for both the training and evaluation phases (**Table 8 and 9**).
>
> - **Q7. Application to other domains involving sequential data.**
>
> 	We believe that the approach of UTR is suitable for sequence optimization problems that require consideration of global input information rather than just local information. For issues such as speech or time-series analysis, we think our solution has potential applicability. More specific applications would require further investigation.

---

> > ### Comment · Reviewer_Edyo · 2024-11-27
> >
> > I appreciate the authors' detailed responses to my questions and concerns. The additional clarifications and experiments have addressed several of my initial concerns, but some points still warrant discussion:
> >
> > 1. Regarding computational overhead and expert models:
> >     * The provided resource/time breakdown for attribute extraction is helpful. However, I remain concerned about the scalability, as the reported processing time for the full dataset is significant.
> >     * While these overheads might be acceptable for offline processing, this should be discussed more explicitly in the paper.
> >
> > 2. On TPL correlation with human judgment: The new human validation experiment shows general alignment between TPL scores and human ratings for both richness and relevance metrics. However:
> >     * The sample size (100 videos) seems relatively small
> >     * It would be helpful to report inter-annotator agreement statistics
> >     * The criteria for human scoring could be more precisely defined
> >
> > 3. I appreciate the addition of failure case analysis in Appendix E.3, particularly regarding knowledge-based questions. This helps better understand the method's limitations.
> >
> > 4. The authors' claim about applicability to other sequential domains (speech, time-series) remains speculative without empirical validation. I would at least write a discussion point in the manuscript about future work.
> >
> > All things considered, I currently maintain my rating of marginally above the acceptance threshold. While the work presents an important contribution in addressing temporal hacking, some concerns about scalability and reproducibility remain. I encourage the authors to:
> > * Further strengthen the empirical validation of TPL.
> > * Provide more detailed analysis of computational costs vs. performance gains and discuss this (even briefly) in their manuscript.
> > * Consider releasing a smaller-scale demonstration of the method to aid reproducibility (since the full implementation is impractical to be reproduced).

---

> > > ### Author Response · Authors · 2024-11-30
> > > **Follow-up the response (Part 1)**
> > >
> > > We sincerely appreciate the reviewer’s further discussions and suggestions. We are willing to continue addressing your relevant concerns. Since new manuscript submissions are now closed, the improvements and clarifications detailed in our response will be implemented in the camera-ready version.
> > > - **Further explanations on scalability from the perspective of resource costs.**
> > > 	This paper primarily demonstrates the scalability of our Video-UTR in terms of data volume and video length, which are two key concerns in the current video MLLM community. Of course, computational resources and time consumption are equally important considerations, but they are not the primary bottlenecks currently limiting the scale-up of video MLLMs. Furthermore, we would like to emphasize that, although constructing UTR-Data requires enough resources and time, these costs can be further optimized. For instance, by employing faster frame extraction techniques and more advanced multi-processing methods, both time and resource consumption can be effectively reduced.
> > > - **More in-depth analysis of the consistency between human judgment and TPL.**
> > > 	We appreciate your further suggestions regarding the human validation experiments for TPL. Specifically,
> > > 	- **The sample size seems relatively small.** We apologize that, due to limitations in time and human resources, we are currently unable to conduct large-scale human evaluation experiments. However, the samples used in our experiment were entirely randomly selected, and the four human annotators involved come from diverse backgrounds (as detailed in the table below). We believe that the final aggregated results are sufficiently convincing.
> > > 		To further address your concerns, we conducted an additional validation experiment with a relatively larger scale (300 cases). As shown in the table below, the results of the two experiments are quite consistent, further supporting the reliability of our validation.
> > >
> > >          | Validation   |Background | Expert Knowledge  |  High          |               | Medium         |               | Low            |               |
> > >       |--------------|--------------| --------------|---------------|---------------|----------------|---------------|----------------|---------------|
> > >       |              | | | **Richness**      | **Relevance**     | **Richness**       | **Relevance**     | **Richness**       | **Relevance**     |
> > >       | TPL level    | | | 3             | 3             | 2              | 2             | 1              | 1             |
> > >       | Human1        | PhD | ✅| 2.79          | 2.86          | 2.02           | 1.88          | 1.57           | 1.66          |
> > >       | Human2        | PhD | ✅ | 2.82          | 2.80          | 2.25           | 1.93          | 1.52           | 1.69          |
> > >       | Human3        | Master| ❌| 2.86          | 2.74          | 2.29           | 1.95          | 1.46           | 1.62          |
> > >       | Human4        | Undergraduate | ❌| 2.77          | 2.84          | 2.16           | 1.87          | 1.32           | 1.56          |
> > >       | Average        |  | |    2.81       |   2.81        |  2.18         |    1.91       |    1.47        |   1.63        |
> > >
> > > 	- **Report inter-annotator agreement statistics.** In the newly added experiments above, we report the background and individual scores of each annotator. We also calulate the Pearson Correlation to quantify this consistent as follows:
> > >
> > >          | Pearson Correlation| Human1 | Human2  | Human3 | Human4 |
> > >       | ----------- | ----------- |----------- | ----------- | ----------- |
> > >       | Human1      |    -    | 0.983       |   0.967     | 0.982|
> > >       | Human2   |  0.983 |  -   |    0.996   | 0.997|
> > >       | Human3 | 0.967 |    0.996   |   -   | 0.992|
> > >       | Human4 | 0.982 |  0.997     |   0.992   | -|
> > > 	- **More precisely defined score setting.** In fact, before the human annotators begin the annotation process, we provide a detailed annotation guideline, which explains the scoring criteria and standards comprehensively and includes relevant references. Specifically, the two composite metrics, *ie.*, richness and relevance, are defined as follows:
> > > 		- **Richness:**
> > > 			- Frame information density, referring to the degree to which each frame corresponds to an independent description.
> > > 			- Level of descriptive detail, referring to the richness of details included from the video.
> > > 			- The richness of motion information, including the extent of motion in both the subject and the scene.
> > > 		- **Relevance:**
> > > 			- The relevance between the video and the text, specifically, the degree to which the description corresponds to the video content.
> > > 			- The relevance of the video context, specifically, the extent to which the descriptions of relevant subjects in the video reflect dynamic changes.
> > >
> > >        Follow the above standards, annotators conducted the validation of the sampled cases and ultimately summarized their assessments into scores for two composite metrics.

---

> > > > ### Author Response · Authors · 2024-11-30
> > > > **Follow-up the response (Part 2)**
> > > >
> > > > - **Add discussion about applicability to other sequential domains in the future work.**
> > > >
> > > > 	Thanks for your suggestions. We discussed to further optimize the data construction pipeline in the existing future work section, including using a single powerful MLLM for attribute extraction. We will take your suggestion into account and add a discussion on the application of UTR in other fields in the camera-ready version. Specifically, we plan to explore the potential applications of temporal hacking in the speech domain, particularly in tasks such as speech synthesis [1]. In this context, systems may attempt to maximize rewards by generating unnatural speech that aligns with target evaluation metrics [2]. For example, if a model is rewarded for producing fluent speech, it might generate sustained tones or repetitive content, neglecting diversity and content accuracy. This highlights the critical importance of constructing appropriate training data and optimization objectives based on the strategies similar to the ***two principles for temporal hacking*** proposed in this study.
> > > >
> > > >   [1] Xu et al., A Survey on Neural Speech Synthesis \
> > > >   [2] RJ et al., Towards End-to-End Prosody Transfer for Expressive Speech Synthesis with Tacotron
> > > >
> > > > - **Provide more detailed analysis of computational costs vs. performance.**
> > > >
> > > > 	We provide an ablation study on the relationship between video data volume and performance in Table 4 of the manuscript. In line with your suggestion, we will add a column to report the time costs, including data construction and training times as follows.
> > > >
> > > >   | Data Scale      | Data Construction Time (Offline) | Training Time (Online)  | MVBench |
> > > >   | ----------- | ----------- |----------- | ----------- |
> > > >   | 0K      |    -    | 22h       | 54.63       |
> > > >   | 180K   | 3.1h  | 23.5h    | 58.45      |
> > > >   | 325K | 5.6h | 25.3h      | 58.78     |
> > > >
> > > > 	All experiments are conducted on 64 NVIDIA A100 GPUs. We can observe that the computational costs associated with scaling up the data at this stage remain within an acceptable range. Therefore, we are currently more focused on the performance improvements it provides.
> > > >
> > > > - **Release a smaller-scale demonstration of the method to aid reproducibility.**
> > > >
> > > > 	Thanks for your valuable suggestions. In fact, we plan to open-source all of the data we have constructed (UTR-Data), allowing researchers to directly use this data to train their models. Additionally, we can also open-source the entire data construction pipeline code for public use. To help reviewer verify its reproducibility, we are providing some data from the construction process for reference. The demo cases sampled from our UTR pipeline including attribute trajectories are presented at <https://huggingface.co/datasets/Submission814/utr_toy> (anonymous link).

---

> > > > > ### Comment · Reviewer_Edyo · 2024-12-02
> > > > >
> > > > > I appreciate the authors' comprehensive responses to my concerns. The additional explanations, empirical validation, and particularly the detailed presentation improvements have strengthened the paper considerably. While some questions about scalability and broader applicability remain, these are reasonable limitations for a conference paper. I am raising my presentation rating to 'good' based on the improved clarity and organization. The overall contribution remains novel and valuable to the field. I maintain my recommendation for acceptance at the current rating tier.

---

### Official Review · Reviewer_4fw2 · 2024-11-04

**Soundness:** 3
**Presentation:** 2
**Contribution:** 3
**Rating:** 6
**Confidence:** 5

**Summary:**

This paper thoroughly analyzes the phenomenon of temporal hacking, where video MLLM learns to spuriously correlate its answer to query prompts to a subset of selected frames without considering the whole video context. The author introduces temporal perplexity (TPL) to assess this misalignment, showing a good correlation with model performance. Moreover, the author proposes the Unhackable Temporal Rewarding (UTR) frameworks and empirically demonstrates their effectiveness by comparing their methods with several open-sourced MLLM.

**Strengths:**

1. The proposed $\mathcal{T}\_{tpl}$ is interesting and correlates well with the model performance.

2. This paper systematically analyzes reward hacking from the RL perspective and proposes two guiding principles for designing unhackable rewards.

3. The proposed method Video-UTR outperforms several baseline methods on both video and image understanding tasks.

**Weaknesses:**

- **Over-complicated Presentation**: I feel the presentation of the paper is unnecessarily complicated and does not explain the details sufficiently. The paper spends too much space introducing the concept of **temporal hacking**, which can be efficiently summarized as *existing training data and objectives make the learned MLLM find shortcuts when answering the query prompt by spuriously correlating the answers with only a subset of video frames, ignoring the whole video frame context.* In contrast, the author is supposed to provide more details of their proposed methods in Section 3.2 instead of putting them in the Appendix.

- **Lack of detailed description of the proposed TPL $\mathcal{T}\_{tpl}$**: Equation 6 defines $\mathcal{T}\_{tpl}$, which requires query the function $\mathcal{R}\_{ppl}$. However, the paper does not provide details about $\mathcal{R}\_{ppl}$ other than citing the original paper developed the $\mathcal{R}\_{ppl}$. For example, does calculating the $\mathcal{R}\_{ppl}$ require access to a specific MLLM?

- **Empirical results**: comparison between the proposed Video-UTR and LLaVA-OneVision is insufficient. The paper only reports LLaVA-OneVision’s results on 3 out of 7 benchmarks, where it outperforms Video-UTR on 2 out of those 3. Besides, the authors should underline PLLaVA's accuracy for ANet-QA rather than underlining their Video-UTR. Additionally, the authors should include results for MLLMs that are also finetuned from QWen2-7B in Table 2. Without these results, it is difficult to assess the source of the performance improvements.

- **Regarding the RL perspective**: Reward hacking is not exclusive to reinforcement learning, as RL is just one approach to maximizing a reward function. In the field of aligning pretrained visual generation models, several methods optimize toward a reward model by directly backpropagating gradients from a differentiable reward model. These approaches are also susceptible to reward hacking, as discussed in [1, 2]. Reward hacking generally occurs when the learning objective fails to fully capture the intended model behavior. I recommend revising this statement and including a citation for [1,2] to provide a broader context.

Minor:

- **Lack of detailed figure captions**: The authors should thoroughly explain each figure within the main text rather than leaving readers to infer their meanings.

- **Missing citations**: Various benchmarks have recently been proposed to evaluate MLLM's video understanding capabilities, e.g., [3, 4]. The authors should consider evaluating their methods on these benchmarks and cite them in the related works sections.

Overall, the presentation of the paper can be significantly improved.

[1] Li et al., Reward Guided Latent Consistency Distillation. TMLR 2024

[2] Zhang et al., Large-scale Reinforcement Learning for Diffusion Models.

[3] Fang et al., MMBench-Video: A Long-Form Multi-Shot Benchmark for Holistic Video Understanding. NeurIPS 2024

[4] He et al., MMWorld: Towards Multi-discipline Multi-faceted World Model Evaluation in Videos.

**Questions:**

1. There is an inconsistency between Table 1 and the text in Line 376 regarding the data scale used for Video-UTR. Table 1 lists the data scale as 1.1M, whereas Line 376 claims the performance is achieved with approximately 700K video data points. Which number is correct?

---

> ### Author Response · Authors · 2024-11-21
> **Response to Reviewer 4fw2**
>
> We greatly appreciate the Reviewer for the recognition of our work and the valuable comments and suggestions you provided. We have responded to each of your comments and questions as follows:
>
> - **Q1. Over-complicated Presentation of temporal hacking and lack of details of UTR.**
>
> 	We greatly appreciate your valuable suggestions. In fact, the aim and key contribution of this work lie in systematically establishing the theory of temporal hacking for video-language modeling from a reinforcement learning (RL) perspective. Therefore, this work devotes significant effort to **introducing, defining, and providing a novel perspective** to interpret the problem of temporal hacking, which we think is more important and valuable than merely proposing a solution to address MLLM shortcut learning. As for the methods section (Section 3.2), we apologize that, due to space constraints, we placed some construction details of UTR in the appendix. In the final version, we will carefully balance the overall layout and pages to make appropriate optimizations.
>
> - **Q2. More detailed description of the proposed TPL.**
>
> 	Thanks for your suggestions. In the latest version, we have added more details about TPL, including the calculation for $R_{ppl}$. For further details, please refer to the **common response** as well as the new version manuscript.
>
> - **Q3. Comparison with LLaVA-OneVision and more comparison results.**
>
> 	The motivation in Video-UTR differs from that of LLaVA-OneVision. LLaVA-OneVision aims to create a general MLLM that incorporates images, videos, and 3D scenes, leading to notable differences in data scale, distribution, and characteristics compared to Video-UTR. Our Video-UTR specifically addresses the issue of temporal hacking in the video domain, which is why we do not perform a comprehensive comparison with LLaVA-OneVision. We believe that the current comparisons are sufficient to demonstrate the effectiveness of our approach. For the image understanding comparison using the same LLM setting, we have included the performance of the LLaVA-NeXT-Video QWen-2 setting in the new version manusrcipt.
>
> - **Q4. Broader statements, more citations about reward hacking and more detailed captions of figure.**
>
> 	Thanks for your valuable suggestions. Following your suggestions, we add more statements and citations about reward hacking and add more detailed figure captions. These improvements are reflected in our latest version. Please refer to the new version manuscript.
>
> - **Q5. More benchmark results.**
>
> 	Thank you for providing these new works and references related to video benchmarks. After carefully reviewing each benchmark, including its open-source status, level of recognition, and impact, we select MMBench-Video [1] to provide additional comparison. The results are shown below.
>
>   | Method      | Overall | Perception | Reasoning |
>   | ----------- | ----------- |----------- | ----------- |
>   | Claude-3.5-Sonnet      | 1.35       | 1.4       |1.04       |
>   | VideoChat2-HD   | 1.23  | 0.44    | 1.23      |
>   | PLLaVA-34B | 1.16 | 1       | 1.1      |
>   | LLaVA-NeXT-Video-34B-HF | 1.13 | 0.58      | 1.03     |
>   | Video-UTR-7B (Ours) | 1.35 | 1.38      | 1.24     |
>
>
>   We can observe that on the latest challenging MMBench-Video benchmark, our Video-UTR remains highly competitive, outperforming numerous 34B and even larger models with its 7B size, demonstrating the superiority of our approach.
>
>   [1] Fang et al., MMBench-Video: A Long-Form Multi-Shot Benchmark for Holistic Video Understanding.
>
>
>
>
>
> - **Q6. Typo error in Line #376.**
>
> 	We greatly appreciate your careful correction. This was indeed a typographical error. The data size indicated in the Table 1 is correct. And we have make this correction in the new version.

---

> > ### Comment · Reviewer_4fw2 · 2024-11-24
> > **Follow-up the discussion**
> >
> > I thank the authors for their detailed response.
> >
> > [Q1] I'm still not sure whether it is a good idea to spend so much space introducing the simple ideas of **temporal hacking**. I'm not saying that temporal hacking is a trivial problem, but it is indeed well-known to the community. Additionally, reward hacking is NOT exclusive to reinforcement learning. In contrast, people will be more interested in **the solutions to the problem** itself. If you plan to balance the overall layout and pages, you should do it now during the rebuttal.
> >
> > [Q3] I believe reporting partial results is not a good practice, as it can lead to incomplete or potentially misleading conclusions. The author's response has not alleviated my concerns and has, in fact, made me more skeptical about the reliability of their results. If you consider the comparison with LLaVA-OneVision to be unfair, then I question why it was included in the first place, as it may give an impression of selective reporting.
> >
> > [Q5] Thank you for including the results.

---

> > > ### Author Response · Authors · 2024-11-27
> > > **Follow-up the response**
> > >
> > > We sincerely appreciate the Reviewer 4fw2 for providing feedback on our response once again.
> > >
> > > **Q1.** We greatly appreciate your further suggestions regarding our writing. Regarding the theoretical framework for temporal hacking established in this paper, we would like to make three additional clarifications.
> > >
> > > 1. This paper serves as the **first to frame video-language modeling as a Markov process and to introduce and define *temporal hacking* based on MDP, offering a novel perspective to the video MLLM community.**
> > > 2. Although reward hacking is not limited to the reinforcement learning (RL) domain, **it originates from RL**, making it appropriate to present the introduction of our temporal hacking from an RL perspective.
> > > 3. Since many current researchers in video MLLM especially video understanding do not have a background in reward hacking, concepts and ideas that may seem straightforward to you might require prior explanation for readers without such expertise.
> > >
> > > Considering the above three points, we think that the prior knowledge and theoretical introduction about temporal hacking are necessary in this paper.
> > >
> > > Regarding the solution details, we have provided the construction process and final examples in Figure 3. In the updated version, we have included more detailed captions, and additional procedural details are provided in the Appendix. Readers who wish to delve deeper into the specific operational steps can refer to these. We believe this arrangement represents a reasonable layout for the current conference paper version.
> > >
> > > **Q3.** We apologize that our explanation in the first round response did not adequately address your concerns. We would like to clarify this further. In fact, **it is not that we selectively reported the results of LLaVA-OneVision [1], rather, the original paper of LLaVA-OneVision only reported results on these specific video benchmarks.**  The other video benchmarks were not evaluated in the original paper. If the reviewer still requires a comprehensive comparison, we can update our reproduced evaluation results accordingly as follows.
> > >
> > > | Methods              | LLM        | Data Scale | MVBench | TempC | VideoMME | MSVD-QA       |          | MSRVTT-QA    |       | TGIF-QA      |       | ANet-QA      |       |
> > > |-----------------------|------------|------------|---------|-------|----------|---------------|----------|--------------|-------|--------------|-------|--------------|-------|
> > > |                       |            |            |         |       |          | **Acc**           | **Score**    | **Acc**          | **Score** | **Acc**          | **Score** | **Acc**          | **Score** |
> > > | LLAVA-OneVision       | Qwen2-7B   | 1.6M       | 56.7    | 59.0  | **58.2**     | 65.3          | 3.8      | 43.3         | 3.0   | 52.8         | 3.4   | **56.6**         | **3.3**   |
> > > | **Video-UTR (Ours)**  | Qwen2-7B   | 1.1M       | **58.8**| **59.7**  | 52.6     | **73.5**      | **4.1**  | **58.3**     | **3.6** | **56.4**    | **3.6** | 55.0    | 3.2|
> > >
> > > [1] Li et al., Llava-OneVision: Easy visual task transfer

---

> > > > ### Comment · Reviewer_4fw2 · 2024-12-03
> > > > **Thank you for the response!**
> > > >
> > > > Regarding Q1, I think it is the author's choice to frame the whole story at the end of the day. I will leave it to the AC to judge this part.
> > > >
> > > > Regarding Q3, the results for the comprehensive comparison address my concern. I hope the author can integrate them into the final manuscript when there is a chance.
> > > >
> > > > Therefore, I raise my confidence to 5 as the paper proposes a solid solution to a notable problem of the current MLLM. I will keep my rating as Weak Accept.

---

### Official Review · Reviewer_PyRQ · 2024-11-06

**Soundness:** 2
**Presentation:** 2
**Contribution:** 3
**Rating:** 6
**Confidence:** 2

**Summary:**

This paper addresses a critical problem in MLLMs for video processing: "temporal hacking," where models simplify the task by focusing on select frames, missing broader context. The authors propose a framework called UTR that mitigates temporal hacking by emphasizing consistent frame activation across the entire video, leading to more accurate video comprehension. This framework incorporates TPL to quantify misalignment in temporal modeling and guides the model through two key principles: high frame information density and inter-frame information dynamics.

**Strengths:**

1.  The paper introduces temporal hacking as a new perspective on reward hacking in RL and effectively adapts this concept to video MLLMs.
2. TPL score is a notable contribution, providing a quantitative way to assess temporal misalignment, a challenge often overlooked in video modeling.
3. The approach shows promise for scalability with additional data, making it versatile for future applications and datasets.
4. The results are good comparing to various larger models.

**Weaknesses:**

1. UTR relies on high-density, high-dynamic frame descriptions. Several datasets is lack of this, I am wondering how are the other methods performance could be improved after simply data modeling.
2. The ablation study is confusing, does table 3 "-Data Modeling" means no data modeling and no task modeling or just no data modeling?
The comparison data scale is different which is unfair to compare.
3. For ablation result, the Video-UTO is not the best without Data Modeling on MMVet and also UTR-Data size of 180K results are sometimes worse than 0K is also confusing.

**Questions:**

1. How is equation 6 used in loss function?
2. For equation 6, why using T:T instead of T-t:T for the second term?
3. Is that possible to get the TPL of your baselines comparing with Video-UTO?

---

> ### Author Response · Authors · 2024-11-21
> **Response to Reviewer PyRQ**
>
> Thank you very much for recognizing our work and for providing such valuable comments and suggestions. We have responded to each of your comments and questions as follows:
> - **Q1. How are the other methods performance could be improved after simply data modeling？**\
> 	To address this question, we would like to present two observations and facts.\
>
> 	1. For existing methods that rely on simple data modeling, such as VideoChat2 [1], adding more video data does not necessarily lead to further performance improvements. This is illustrated in both **Figure 1 (a)** and **Table 3** in our manuscript.
> 	2. To further substantiate this point, we conducted comparative experiments on WebVid, where we trained models using both a randomly sampled set of 1 million WebVid videos and a 1K subset sampled based on our proposed TPL score. As shown in the following table:
>
>         | Training Data  | Size      | MVBench |
>         | ----------- | ----------- | ----------- |
>         | Random Sample      | 1M       | 50.6 |
>         | TPL Sample  | 1K        | 51.2 |
>
> 		The model trained with the 1K WebVid samples outperforms the one trained on the randomly sampled 1 million videos. This finding underscores that with simple video data modeling, increasing data volume may actually lead to performance degradation, aligning with the ***“anti-scaling law”*** phenomenon we discuss in our paper.
> 	3. Existing video understanding benchmarks, such as VideoMME [3], are not optimally designed. Many test cases (shown in **Figure 9** in Appendix) in these benchmarks can be answered correctly by observing only a single frame or a few frames and then relying on the LLM’s inherent knowledge (language priors) for reasoning. This design biases current video data modeling approaches towards equipping models with more knowledge rather than enhancing their ability to observe and understand video content comprehensively.
> 	4. Some recent studies have recognized the issue in existing benchmarks and made adjustments. For example, Tarsier [1] increases data diversity and reduces low-quality data that may introduce shortcuts by filtering verbs from captions using the model. COSMO [2] applies GRiT to obtain detailed annotation as prompts for GPT-4 to generate high-quality video captions. Merlin [3], on the other hand, guides the model to better capture spatiotemporal cues by constructing trajectory chain-of-thought data. These methods, to some extent, align with the two principles we proposed.
>
> 	[1] Wang et al., Tarsier: Recipes for Training and Evaluating Large Video Description Models.\
> 	[2] Wang et al., COSMO: COntrastive Streamlined MultimOdal Model with Interleaved Pre-Training.\
> 	[3] Yu et al., Merlin: Empowering multimodal llms with foresight minds.
>
> - **Q2. Further explanation of the ablation study in Table 3.**
>
> 	We apologize for any confusion caused by the description of the settings in the ablation study in **Table 3**. In fact, we used a progressive ablation approach, gradually removing components we designed from the final model, Video-UTR. Therefore, the third row, labeled **“- Data Modeling”**，indicates the removal of both data modeling and task modeling. To avoid unfair comparisons due to reduced data volume, we added an equivalent amount of video data in the **row 4** based on the **row 3** settings. As shown in the results, the performance remains lower than that of our base method.
>
> - **Q3. The metrics for several datasets (MMBench, VideoMME) in Table 3 and Table 4 are not the best.**
>
> 	MMVet, as an image-level understanding benchmark, is not highly sensitive to video data, and we just want to provide an additional reference. VideoMME, as explained in **Q1**, contains many evaluation cases that test the knowledge base of the LLM itself rather than its ability to understand spatiotemporal contexts, which is why our method do not yield a significant improvement.
>
> - **Q4. How is equation 6 used in loss function?**
>
> 	In the practical application of Eq.6, TPL is not incorporated as a loss in the training process. Instead, it is applied in an offline manner to reflect the temporal perplexity score of the model in video-language modeling. This score serves as a metric to assess misalignment in temporal modeling.
>
> - **Q5. For Eq.6, why using T:T instead of T-t:T for the second term?**
>
> 	In fact, the definition of Eq.6 corresponds to Eq.5. We explained the reason for this in Line #205 of the manuscript, namely that **“To model an extreme case of proxy cumulative reward, we use a single, *i.e.*, k = t, randomly sampled keyframe to represent the entire video context”**. Here $t = T$ and **T:T** means the single frame we selected.
>
> - **Q6. Is that possible to get the TPL of baselines comparing with Video-UTO?**
>
> 	Yes, it is. In fact, we have already shown the comparison of TPL scores between the baseline (VideoChat2) and Video-UTR (UTR-Data) in Figure 2(a) in the main text.

---

> > ### Author Response · Authors · 2024-12-03
> > **Kind Reminder to Reviewer PyRQ**
> >
> > Dear Reviewer PyRQ,
> >
> >   We sincerely thank you for your valuable comments and suggestions on our work during this review process. We hope that our responses, including our individual reply to your feedback, the common response, and the responses to other reviewers, have addressed your concerns and questions. As the author rebuttal phase is coming to an end, we would like to kindly remind you to let us know if there are any remaining questions or issues you would like us to clarify further. We greatly value your insights and are more than happy to provide any further clarifications if needed to address any outstanding concerns.
> >
> > If your concerns have already been resolved, we would greatly appreciate your consideration of a higher rating, as this would play a significant role in the final evaluation of our work. Thank you again for your time and support!
> >
> > Best,\
> > Authors

---

> > > ### Comment · Reviewer_PyRQ · 2024-12-03
> > >
> > > Dear Authors,
> > >
> > > Thank you for your clarifications. My questions were fully addressed and I have adjusted my scores accordingly.

---

### Author Response · Authors · 2024-11-21
**Common Response**

**Common response:**

First of all, we thank all the Reviewers for their insightful and valuable comments. We are encouraged that the reviewers all give the high praise to our work including **novel perspective** (Reviewer PyRQ, 4fw2, Edyo), **clear theoretical foundation** (Reviewer PyRQ, 4fw2, Edyo, ezXT), **practical metric** (Reviewer 4fw2, Edyo), **effective methods** (Reviewer PyRQ, 4fw2, Edyo, ezXT), and **promising experiments** (Reviewer PyRQ, 4fw2, Edyo, ezXT).

And then, we sincerely thank all Reviewers for their valuable suggestions and questions. Due to the presence of several common or similar issues and suggestions, we provide a unified response here.

- **Updates to the Manuscript**. Taking into account the reviewers’ comments and suggestions for this work, we have made the following updates to the manuscript.
	1. Add more detailed captions for each figure.
	2. Add more detailed explanation of the proposed $\mathcal{T}_{tpl}$. (Line #212-#218).
	3. Add more comparison methods. (Table 2).
	4. Add more statements and citations about reward hacking. (Line #524-#528).
	5. Add more experimental details about Video-UTR. (Appendix B).
	5. Add more discussions about expert models, failure cases and $\mathcal{T}_{tpl}$. (Appendix E).
	6. Correct some statements and typographical errors.

	All changes in the manuscript are highlighted in blue.

- **Clarification Regarding the UTR Method**. In this work, we propose a *unhackable temporal rewarding (**UTR**)* framework for video MLLM to mitigate the temporal hacking. UTR contains two stages including *data modeling* and *task modeling*. Data modeling employs multiple expert models to extract spatiotemporal attributes, thereby constructing attribute trajectories. Task modeling, in turn, builds upon these attribute trajectories by utilizing a bidirectional query mechanism to formulate optimization objectives. **Notably**, **the entire UTR process, including data modeling and task modeling, is conducted offline and does not interfere with the training process.** As a result, **it does not impose any additional computational burden on the model.** Essentially, UTR is a general data construction framework, akin to Tarsier[1] and Momentor[2], that can be seamlessly integrated into existing video MLLMs.

	[1] Wang et al., Tarsier: Recipes for Training and Evaluating Large Video Description Models.\
	[2] Qian et al., Momentor: Advancing video large language model with fine-grained temporal reasoning.


- **More Discussions about Expert Model**. The use of expert models is a critical component of UTR. To help readers better understand this aspect and address some concerns raised by reviewers, we have added a dedicated section in the **Appendix E.1** of the manuscript discussing expert models. Specifically, we focus on the following topics: the importance and rationale behind selecting expert models for attribute extraction, the extraction and filtering of high-quality attributes, and human validation of the extracted attributes. For detailed information, please refer to the content in the Manuscript.

Finally, we have provided responses to each comment and question from the each reviewer, hoping to address your concerns adequately.

---

### Meta-Review · Area_Chair_mBhs · 2024-12-18

**Metareview:**

The paper introduces Unhackable Temporal Rewarding (UTR), a framework addressing "temporal hacking" in video MLLMs where models exploit shortcuts by focusing on select frames rather than understanding full video narratives. This phenomenon contributes to an anti-scaling paradox where increased data and model size lead to degraded performance.

All reviewers praised the paper's novel perspective on framing temporal hacking through a reinforcement learning lens, the introduction of the Temporal Perplexity (TPL) metric for assessing temporal misalignment, and strong empirical results demonstrating UTR's effectiveness. The theoretical foundation connecting reward hacking to video understanding was considered particularly insightful.

Initial concerns focused on presentation clarity, computational overhead of expert models, reproducibility, and selective reporting of baseline comparisons. The authors provided detailed responses addressing these issues, including comprehensive human validation experiments for TPL, detailed computational cost breakdowns, and complete comparison results across all benchmarks. Two reviewers explicitly noted satisfaction with these clarifications, with one raising their presentation score.

While some reservations remain about scalability and broader applicability beyond video understanding, reviewers agreed these were reasonable limitations for a conference paper. The consensus is that UTR represents a valuable contribution by systematically addressing a fundamental challenge in video MLLMs.

**Additional Comments On Reviewer Discussion:**

None -- see metareview.

---

### Decision · Program_Chairs · 2025-01-22

Accept (Poster)